# *Yersinia* remodels epigenetic histone modifications in human macrophages

**Indra Bekere**[1]*, **Jiabin Huang**[1], **Marie Schnapp**[1], **Maren Rudolph**[1], **Laura Berneking**[1], **Klaus Ruckdeschel**[1], **Adam Grundhoff**[2], **Thomas Günther**[2], **Nicole Fischer**[1], **Martin Aepfelbacher**[1]*

1 Institute of Medical Microbiology, Virology and Hygiene, University Medical Center Hamburg-Eppendorf (UKE), Hamburg, Germany, 2 Heinrich-Pette-Institute (HPI), Leibniz Institute for Experimental Virology, Research Group Virus Genomics, Hamburg, Germany

* ibekere@uke.de (IB); m.aepfelbacher@uke.de (MA)

**Data Availability Statement:** ChIP-seq data have been deposited in the ArrayExpress database at EMBL-EBI (www.ebi.ac.uk/arrayexpress) under accession number E-MTAB-10475. RNA-seq data

## Abstract

Various pathogens systematically reprogram gene expression in macrophages, but the underlying mechanisms are largely unknown. We investigated whether the enteropathogen *Yersinia enterocolitica* alters chromatin states to reprogram gene expression in primary human macrophages. Genome-wide chromatin immunoprecipitation (ChIP) seq analyses showed that pathogen-associated molecular patterns (PAMPs) induced up- or down-regulation of histone modifications (HMod) at approximately 14500 loci in promoters and enhancers. Effectors of *Y. enterocolitica* reorganized about half of these dynamic HMod, with the effector YopP being responsible for about half of these modulatory activities. The reorganized HMod were associated with genes involved in immune response and metabolism. Remarkably, the altered HMod also associated with 61% of all 534 known Rho GTPase pathway genes, revealing a new level in Rho GTPase regulation and a new aspect of bacterial pathogenicity. Changes in HMod were associated to varying degrees with corresponding gene expression, e. g. depending on chromatin localization and cooperation of the HMod. In summary, infection with *Y. enterocolitica* remodels HMod in human macrophages to modulate key gene expression programs of the innate immune response.

## Author summary

Human pathogenic bacteria can affect epigenetic histone modifications to modulate gene expression in host cells. However, a systems biology analysis of this bacterial virulence mechanism in immune cells has not been performed. Here we analyzed genome-wide epigenetic histone modifications and associated gene expression changes in primary human macrophages infected with enteropathogenic *Yersinia enterocolitica*. We demonstrate that *Yersinia* virulence factors extensively modulate histone modifications and associated gene expression triggered by the pathogen-associated molecular patterns (PAMPs) of the bacteria. The epigenetically modulated genes are involved in several key pathways of the macrophage immune response, including the Rho GTPase pathway, revealing a novel level of Rho GTPase regulation by a bacterial pathogen. Overall, our findings provide an in-depth

have been deposited in the ArrayExpress database at EMBL-EBI (www.ebi.ac.uk/arrayexpress) under accession number E-MTAB-10473. All remaining relevant data are within the manuscript and its Supporting Information files.

**Funding:** The investigators IB, AG, TG, NF and MAe were supported by the research consortium EPILOG, funded by the Stiftung zur Förderung der wissenschaftlichen Forschung in Hamburg (LFF-FV44: EPILOG) https://h-w-s.org/. The investigators MS, JH, MR, LB and KR were supported by Universitätsklinikum Hamburg-Eppendorf https://www.uke.de. The funders had no role in study design, data collection and analysis, decision to publish, or preparation of the manuscript.

**Competing interests:** The authors have declared that no competing interests exist.

view of epigenetic and gene expression changes during host-pathogen interaction and might have further implications for understanding of the innate immune memory in macrophages.

## Introduction

Macrophages play an essential role in the response to bacterial infection. They sense pathogen-associated molecular patterns (PAMPs) like lipopolysaccharide (LPS), nucleic acids or flagellin through Toll-like, RIG-I-like or NOD-like pattern recognition receptors (PRRs), respectively [1,2]. RIG-I-like and NOD-like PRRs are part of inflammasomes that process and release the major pro-inflammatory cytokines interleukin (IL)-1β and IL-18 [3,4]. While PRRs recognize a wide variety of PAMPs, their downstream signaling often converges on mitogen activated protein kinase- (MAPK), nuclear factor κB- (NF-κB) and type I interferon (IFN) signal pathways [2,5]. These pathways include activation of transcription factors that control expression of genes for cytokines, chemokines, and inflammasome components, as well as genes for metabolism, cytoskeleton regulation, and transcriptional regulation [2,6–9]. Thousands of functionally diverse genes are up- or downregulated by PAMPs in macrophages. Many of these genes belong to elaborate transcriptional programs that drive macrophage functions during infection. Such programs contribute crucially to the immunological phenomena of priming, tolerance (immunosuppression) and trained immunity [10,11]. The inflammatory gene expression programs in macrophages form an intricate network that is characterized by cross-talk and feedback loops [10,12]. Because of their central role in immune defense, numerous pathogenic bacteria have developed mechanisms to suppress or modulate macrophage gene expression [13].

Pathogenic *Yersinia* species, which comprise the enteropathogens *Y. pseudotuberculosis* and *Y. enterocolitica* as well as the plague agent *Yersinia pestis*, can proliferate extracellularly in lymphoid tissues of animal hosts [14,15]. Yersiniae can on one hand exploit intracellular niches early during infection, e.g., within macrophages and dendritic cells, in order to disseminate to lymphoid tissues and organs such as spleen and liver [16–20]. On the other hand these bacteria have been shown to suppress phagocytosis, migration and immune signaling in resident cells of the innate immune system [15,21]. The major virulence mechanism of pathogenic yersiniae is a type III secretion system (T3SS) by which they inject effector proteins, named *Yersinia* outer proteins (Yops), into immune cells [15,22,23]. The seven known Yops exert different activities to suppress immune cell functions. E.g., YopE, YopT and YopO block cytoskeletal reorganization by modifying the activity of Rho GTP binding proteins [24,25]. Further, YopP/J, and YopM inhibit the inflammatory responses triggered by the PAMPs (e.g., LPS, heptose containing metabolites, T3SS pore or the adhesins YadA and invasin) or the effector-triggered immunity (ETI) elicited by the bacteria [26–34]. Induction of ETI is a consequence of bacterial virulence activities in host cells that disrupt cellular processes and are sensed by the host [34]. *Yersinia* triggers ETI e.g., by the T3SS translocation pore-produced membrane damage or by deactivation of Rho GTPases through YopE and YopT [29]. YopP/J acetylates and inhibits components of NF-κB and MAPK pathways and thereby profoundly suppresses pro-inflammatory gene expression downstream of TLRs [28]. YopM acts by both increasing production of the anti-inflammatory cytokine IL-10 and counteracting activation of the pyrin inflammasome, which is triggered by the Yop-induced deactivation of Rho GTPases [27,35,36]. Thus, on one hand pathogenic yersiniae contain immunostimulatory PAMPs and their T3SS effectors elicit ETI that strongly alter gene expression in macrophages.

On the other hand, the bacteria employ an arsenal of activities to systematically modulate and antagonize the PAMP- and ETI triggered immune responses [29,34,37].

Epigenetic mechanisms play a master role in the regulation of macrophage gene expression. Among others they i) determine the phenotypes of macrophages in different tissues and disease states, ii) control trained innate immunity, iii) regulate priming and immune tolerance and iv) integrate stimulus triggered responses [10,11,38,39]. Gene expression in response to external stimuli is controlled by the concerted activity and binding of macrophage lineage-determining transcription factors and stimulus-regulated transcription factors to accessible cis-regulatory elements (promoters and enhancers) in the cellular chromatin [38,39]. DNA accessibility is determined by the pre-existing epigenomic landscape, which is the sum of DNA methylation, nucleosome occupancy, pre-bound factors and histone modifications [40]. The types of histone modifications located at gene promoters and enhancers influence DNA accessibility and transcriptional activity. For instance, a trimethylated lysine at position 4 in histone 3 (H3K4me3) is characteristic for active promoters, an acetylated lysine at position 27 in histone 3 (H3K27ac) indicates active promoters and enhancers while H3K9me3 signifies repressed, transcriptionally inactive heterochromatin[41–43]. Most posttranslational histone modifications are transient and their turnover is regulated by enzymes that mediate their deposition or deletion [44,45]. Bacterial PAMP-induced signaling has been shown to intensely modulate histone modifications and this has been implicated in priming, immune tolerance and trained immunity in macrophages [10,11,38,46]. Consequently, numerous pathogens can interfere with histone modifications in host cells using unique and sophisticated strategies [47–49].

The current understanding of how *Yersinia* modulates gene expression in macrophages is that it uses its effectors YopP and YopM to interfere with PAMP-induced MAP-kinase and NF-kB signaling to the nucleus [29]. Previous microarray or RNA-seq studies of *Yersinia* effects on gene expression were so far conducted in *Y. pestis* infected rat lymph nodes [50], African green monkeys [51,52] and neutrophils [53], in *Y. pseudotuberculosis* infected mouse cecum [54] and Peyer's patches [55] as well as in *Y. enterocolitica* infected epithelial and NK cells [56,57]. In macrophages systematic gene expression analysis was performed during *Y. enterocolitica* or *Y. pseudotuberculosis* infection of cultured mouse macrophages for up to 4 h and utilizing microarrays [30,58–60]. Here we employed primary human macrophages and genome wide chromatin immunoprecipitation (ChIP)-seq and RNA-seq technologies to globally analyze histone modifications and associated gene expression patterns for up to 6 h of *Y. enterocolitica* infection. We uncover a profound and coordinated reorganization of histone modifications as basis for systematic *Yersinia* effects on gene expression in macrophages. In addition to extensive modulation of histone modifications at inflammatory, immune response and metabolism genes, we found that the bacteria caused alterations of histone modifications at 61% of all 534 known Rho GTPase pathway genes in macrophages. These results describe the previously unrecognized strategy of pathogenic *Yersinia* to modulate specific gene expression programs in innate immune cells through the systematic reorganization of histone modifications.

## Results

### Global reorganization of histone modifications in macrophages upon *Yersinia enterocolitica* infection

We first investigated on a global scale whether *Y. enterocolitica* alters histone modification patterns in human macrophages [38]. For this, *in vitro* differentiated primary human macrophages were mock infected or infected with *Y. enterocolitica* wild type strain WA314 or its

avirulent derivative WAC and subjected to chromatin immunoprecipitation (ChIP)-seq (Fig 1A). WAC lacks a T3SS and therefore was used to separate the effects of the *Yersinia* PAMPs from the T3SS associated effects (for strains employed in this study see S1 Table). We investigated cells after infection with a multiplicity of infection (MOI) 100 for 6 h, conditions at which 100% of cells were positive for translocated Yops (S1A Fig) and a maximal effect on gene expression [36] and histone modifications (S1B Fig) but no signs of cell death as measured by membrane permeability (S1C Fig) and caspase-3 cleavage (S1D Fig) [61,62] are detectable in the human macrophages.

ChIP-seq analyzes were performed for four histone-3 (H3) marks whose impact on macrophage gene expression has been well established: H3K4me3, indicating active promoters; H3K4me1, which is highly enriched at distal regulatory elements (enhancers); H3K27ac, indicating active promoters and enhancers; and H3K27me3 indicating inactive promoters and enhancers [41–43,63]. Successful ChIP was confirmed using ChIP-qPCR with positive and negative control primers for different histone marks (S2A Fig). Dynamic regions were defined as those exhibiting at least a 2-fold change in any pairwise comparison between mock-, WA314- or WAC infected macrophages. Overall, H3K27ac peaks were the most dynamic (43%) followed by H3K4me3 peaks (7%), H3K4me1 regions (3%) and H3K27me3 regions (0.1%) (Fig 1B). Around half of the dynamic H3K4me3 peaks and H3K4me1 regions and one fourth of the dynamic H3K27ac peaks were at promoters (+/- 2 kb from transcription start site; Fig 1C). The remaining half of the dynamic H3K4me3 peaks and H3K4me1 regions and three fourth of the dynamic H3K27ac peaks were at enhancers (H3K4me1-enriched regions outside promoters; Fig 1C).

A Spearman correlation heatmap of the H3K27ac peaks and two corresponding datasets published elsewhere from naive and *E. coli* LPS-stimulated primary human macrophages [10,64] revealed i) a strong correlation between WAC infected and LPS-treated macrophages (Fig 1D), indicating that the PAMP-induced histone modifications are to a large part caused by LPS, although yersiniae possess a number of additional PAMPs [26,65,66]; ii) a close similarity between the mock-infected macrophages in our study and the naïve macrophages employed in other studies, and iii) a higher similarity of WA314 infected macrophages to naïve and mock-infected macrophages than to WAC- and LPS infected macrophages (Fig 1D). This likely reflects suppression of PAMP/LPS-induced H3K27ac modifications by the T3SS effectors of WA314 [29].

We evaluated whether histone marks were significantly ($\geq$ 2-fold change, adjusted P-value $\leq$ 0.05) up- or down-regulated in the three pairwise comparisons WAC vs mock, WA314 vs mock and WAC vs WA314 (Fig 1E and S18 Table). An eminent result was that, consistently, around 14-times more H3K27ac regions than H3K4me3 regions were dynamic (Fig 1E and S18 Table). Further, 2156 unique H3K4me1 regions were dynamic whereas H3K27me3 regions were essentially unaltered under these conditions (Fig 1E and S18 Table), further suggesting that H3K27me3 marks are not affected by *Yersinia* infection in macrophages. WAC up- and down-regulated histone modifications at 14559 unique loci vs mock, reflecting the effect of the PAMPs (Fig 1E). As determined by comparison with the WAC effects, WA314 inhibited the up regulation of 571 H3K4me3 regions (53%; Fig 1F) and 2881 H3K27ac regions (42%; Fig 1G) and inhibited the down regulation of 2627 H3K27ac regions (40%; Fig 1G).

Further analysis revealed that histone modification patterns were modified at promoters of 6228 genes and at enhancers of 7730 genes, whereby 2964 genes were modified both at the promoter and nearby enhancer regions (Fig 1H). We conclude that *Y. enterocolitica* triggers extensive reprogramming of chromatin states in human macrophages by regulating H3K27ac-, H3K4me1- and H3K4me3 marks at promoters and enhancers of about 11000 genes (Fig 1H).

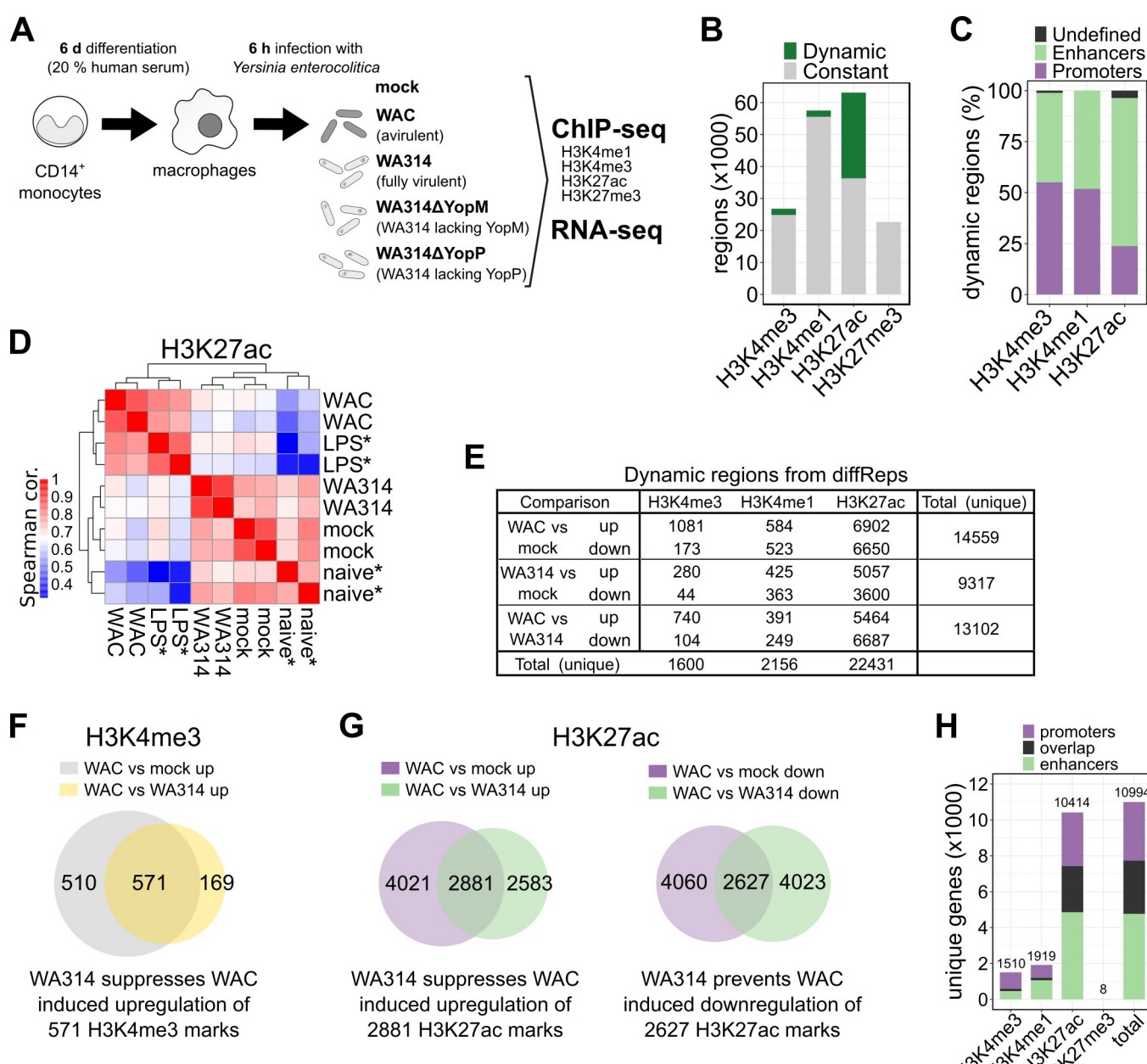

**Fig 1. Changes of epigenetic histone modifications in *Y. enterocolitica* infected primary human macrophages. A,** Experimental design. CD14[+] monocytes were isolated and differentiated into macrophages by cultivation with 20% human serum for 6 days. Macrophages from ≥ two independent donors were mock infected or infected with avirulent *Y. enterocolitica* strain WAC, wild type strain WA314 or the single Yop-mutant strains WA314ΔYopM and WA314ΔYopP with multiplicity-of-infection (MOI) of 100 for 6 h. Samples were subjected to ChIP-seq for histone modifications H3K27ac, H3K4me3 and RNA-seq. H3K4me1 and H3K27me3 ChIP-seq was performed only for mock, WAC and WA314 infected macrophages. For all replicates used in the analysis see S17 Table. **B,** Bar plot showing proportions of dynamic and constant regions of ChIP-seq data obtained from mock-, WAC- and WA314 infected macrophages as described in (**A**). Dynamic regions were defined as MACS (H3K4me3, H3K27ac) or SICER (H3K4me1, H3K27me3) peaks overlapping significantly (≥ 2-fold change, adjusted P-value ≤ 0.05) up- or down-regulated differentially enriched regions from diffReps in the three pairwise comparisons WAC vs mock, WA314 vs mock and WAC vs WA314. **C,** Bar plot showing distribution of dynamic regions from (**B**) at gene promoters and enhancers. **D,** Heatmap showing Spearman correlation (cor.) of H3K27ac tag density at all H3K27ac peaks from mock-, WAC- and WA314-infected samples and publicly available H3K27ac ChIP-seq data of naive* and LPS* treated macrophages. Low to high correlation is indicated by blue-white-red colour scale. **E,** Number of differentially enriched regions (diffReps ≥ 2-fold change, adjusted P-value ≤ 0.05) for the indicated histone marks. Total (unique) refers to all unique regions in each row or column after merging overlapping regions. **F, G,** Venn diagrams showing overlaps of differentially enriched regions from diffReps depicting suppression of WAC induced upregulation of H3K4me3 marks by WA314 (**F**) and suppression of WAC induced up-regulation of H3K27ac marks by WA314 or prevention of WAC induced down-regulation of H3K27ac marks by WA314 (**G**). **H,** Bar plot showing the number of genes associated with promoters, enhancers or both for the indicated histone marks associated with dynamic regions in (**E**) or when taking all regions together (total).

## *Y. enterocolitica* virulence factors suppress PAMP-induced histone modifications at promoters and enhancers

We investigated the bacteria induced reprogramming of histone modifications at promoters and enhancers in more detail. By genome wide analysis of dynamic H3K27ac and H3K4me3 regions at promoters of mock-, WAC- and WA314 infected macrophages we identified 10 clusters which could be assigned to 6 classes and 2 modules (Methods; Fig 2A and S2 Table). Promoter module 1 (P1) contains dynamic H3K4me3- and H3K27ac regions that change concordantly (classes P1a, b; Fig 2A and 2B). Promoter module 2 (P2) contains dynamic H3K27ac regions at largely constant H3K4me3 regions (classes P2a-d; Fig 2A and 2B). The patterns of histone modifications in the classes were clearly distinct and assigned to 4 different profiles. In profile "Suppression", WAC increases deposition of histone marks and this is suppressed by WA314 (P1a and P2a; Fig 2A–2C). In profile "Prevention", WAC downregulates histone marks and this is prevented by WA314 (P1b and P2d; Fig 2A–2C). In profile "Down", WA314 selectively downregulates H3K27ac marks when compared to mock and WAC (P2b; Fig 2A–2C). In profile "Up" WA314 selectively upregulates H3K27ac marks when compared to mock and WAC (P2c; Fig 2A–2C). These profiles are also illustrated by tag densities of H3K4me3 and H3K27ac ChIP-seq at promoters of selected genes (Fig 2C). WA314 strongly counteracted the PAMP-induced up- and downregulation of histone modifications at promoters in Suppression and Prevention profiles, which was also confirmed by ChIP-qPCR analysis of H3K4me3 and H3K27ac for IL6, IL1B and PTGS2 (Suppression) and ASB2 and CEBPE (Prevention) genes (S2B Fig). However, WA314 counterregulation often was not complete therefore the levels of histone modifications in WA314 infected cells were frequently in between the levels in mock- and WAC infected macrophages (e.g. in Suppression profile classes P1a and P2a; Fig 2A and 2B).

We further sought to investigate in detail the *Yersinia* induced reprogramming of histone modifications at distal regulatory elements/enhancers. A genome wide heatmap of dynamic H3K27ac regions at enhancer regions, characterized by the presence of H3K4me1 marks, was prepared in mock-, WAC- and WA314 infected macrophages (Fig 2D and S3 Table). Six clusters comprising altogether 16408 dynamic enhancers were identified and assembled into four classes (E1-E4; Fig 2D). Analogous to the promoter classes, the enhancer classes E1, E2, E3 and E4 correspond to Suppression, Down, Up and Prevention profiles, respectively (Fig 2E). In unstimulated macrophages (mock) enhancers were either poised (no or low level of H3K27ac marks; E1 and E3) or constitutive (presence of H3K27ac marks; E2 and E4) (Fig 2D and 2E) [67]. WAC infection activated poised enhancers by increasing H3K27ac marks (E1) or repressed constitutive enhancers by decreasing H3K27ac marks (E4) (Fig 2D and 2E). WA314 inhibited the WAC induced up-regulation of poised enhancers in Suppression profile E1 and down-regulation of constitutive enhancers in Prevention profile E4 (Fig 2D and 2E). WA314 also down-regulated constitutive enhancers in Down profile E2 and up-regulated poised enhancers in Up profile E3 (Fig 2D and 2E).

We also investigated the effect of *Yersinia* on latent enhancers, defined by the initial absence of H3K4me1- and H3K27ac marks [67]. We identified 149 latent enhancers in mock infected macrophages that gained H3K4me1- and H3K27ac marks upon WAC infection, which converts these latent enhancers into a constitutive state (Fig 2F and S4 Table). WA314 also increased H3K4me1 levels but did not produce an increase of H3K27ac levels in most of these latent enhancers (Fig 2F). Thus, WA314 infection does not inhibit the unveiling of latent enhancers and their conversion into a poised state by the bacterial PAMPs, but suppresses their transition to a constitutive state.

In summary, four profiles describe how *Y. enterocolitica* alters H3K4me3 and H3K27ac marks at macrophage promoters and enhancers. In the profiles Suppression and Prevention,

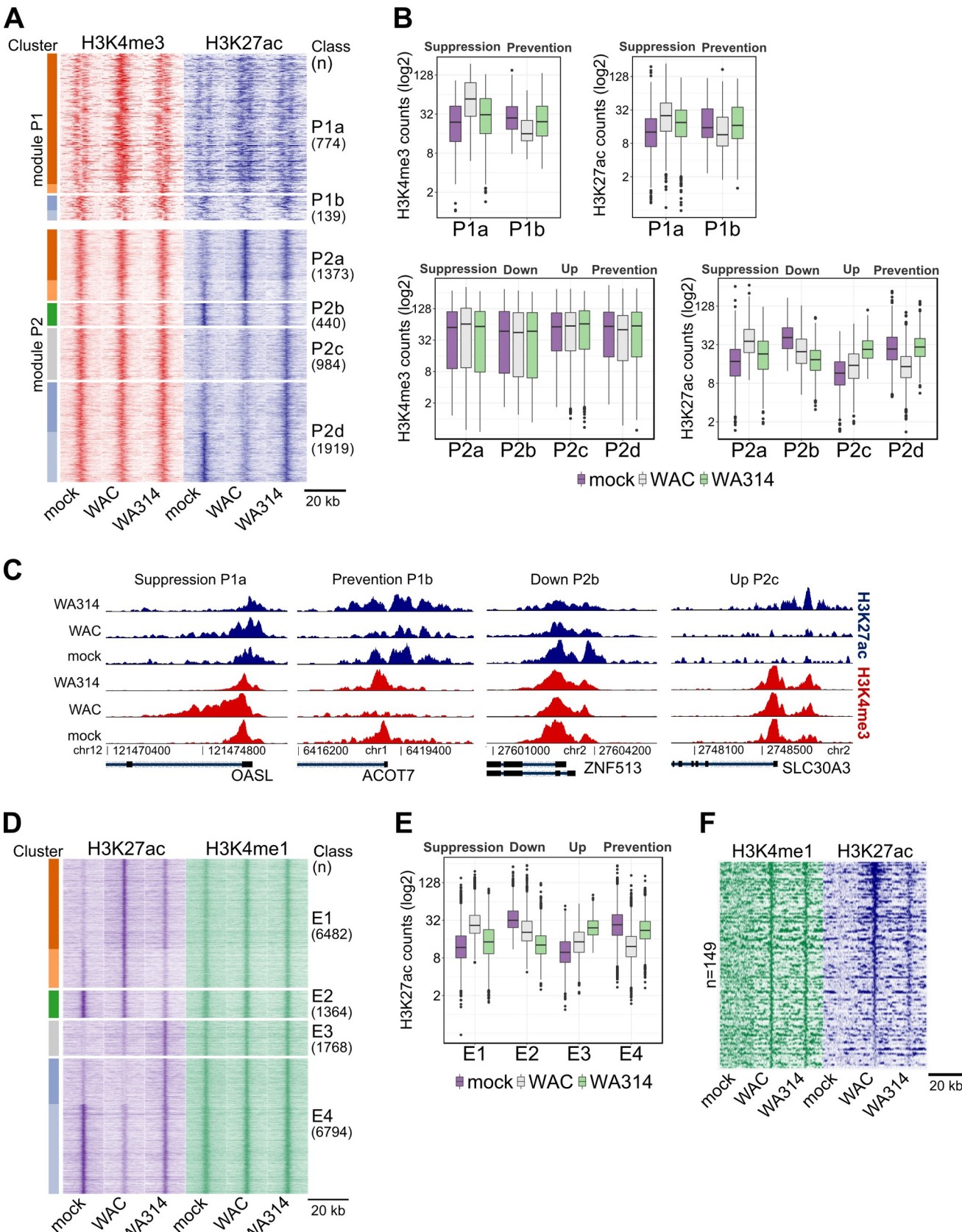

**Fig 2. Dynamic chromatin modifications at promoters and enhancers in *Y. enterocolitica* infected macrophages. A,** Heatmap showing clustering of all H3K4me3 (module P1) and H3K27ac (module P2) differentially enriched regions at promoters in mock-, WAC- and WA314 infected human macrophages. P1 contains dynamic H3K4me3 regions and P2 contains dynamic H3K27ac regions at largely constant H3K4me3 regions. H3K4me3 and H3K27ac tag densities are shown in both modules. The identified clusters (colour coded bars on the left side) were grouped in classes P1a-P2d. **B,** Boxplots of H3K4me3 and H3K27ac tag counts for the classes in (**A**). Profiles describing relation of histone mark levels between mock, WAC and WA314 are indicated on top. Data are representative of at least two independent experiments. **C,** Peak tracks of H3K4me3 (red) and H3K27ac (blue) ChIP-seq tag densities at promoter regions of the genes OASL from Suppression profile classes P1a, ACOT7 from Prevention profile class P1b, ZNF513 from Down profile class P2b and SLC30A3 from Up profile class P2c. **D,** Heatmap showing clustering of H3K27ac differentially enriched regions at enhancers of mock-, WAC- and WA314 infected macrophages. H3K4me1 tag counts are shown for the associated regions. Clustering analysis yielded 6 clusters (color coded, left side) which were assembled into classes E1-4. **E,** Boxplot of H3K27ac tag counts of classes E1-E4 in (**D**). Profiles describing relation of histone mark levels between mock, WAC and WA314 are indicated on top. Data are representative of two independent experiments. **F,** Heatmap showing H3K4me1 and H3K27ac tag counts at latent enhancers induced by WAC vs mock. For (**A**), (**D**) and (**F**) rows are genomic regions from -10 to +10 kb around the centre of the analyzed regions. "n" indicates number of regions.

the bacteria's virulence plasmid-encoded factors block the PAMP-induced deposition or removal of H3K4me3- and H3K27ac marks, respectively. In profiles Down and Up the T3SS-associated effectors down- or up-regulate, respectively, H3K27ac marks.

## *Yersinia* reorganization of histone marks and gene expression targets central immune pathways in macrophages

We asked to what extent the histone modifications reprogrammed by the bacteria associate with expression of the corresponding genes. Therefore, infected macrophages were analyzed by RNA-seq in parallel to ChIP-seq (Fig 1A). 6148 differentially expressed genes (DEGs) were altogether found in three pairwise comparisons (WAC vs mock, WA314 vs mock and WAC vs WA314; Fig 3A and S5 and S6 Tables). Four classes (R1-R4) were identified in the DEGs which corresponded to Suppression- (R1), Down- (R2), Up- (R3) or Prevention (R4) profiles (Fig 3A and S6 Table). WAC up-regulated 3020 genes (Fig 3B) and down-regulated 2152 genes (Fig 3C) vs mock reflecting the transcriptional response of macrophages to the PAMPs of *Y. enterocolitica*. In the Suppression profile, 42% of the genes upregulated by WAC were not or only partly upregulated by WA314 (Fig 3B). In the Prevention profile, 39% of the genes downregulated by WAC were not or much less downregulated by WA314 (Fig 3C). RT-qPCR analysis confirmed WA314 counteraction of PAMP-induced changes in mRNA levels for PPARGC1B (Prevention) and ABI1, IL1B and TNF (Suppression) genes (S2C Fig). WA314 often did not completely reverse the PAMP-induced up- or downregulation of genes, as shown by the fact that gene expression levels in the WA314 infected cells were frequently in between mock- and WAC infected macrophages (e.g. in classes R1 and R4, Fig 3A). Thus, coinciding with histone modifications, gene expression is modulated extensively by *Y. entero-colitica* in human macrophages and can also be divided into Suppression-, Down-, Up-, and Prevention profiles.

We evaluated all individual histone modifications which associated with gene expression changes in the different profiles and found that Suppression-, Down-, Up- and Prevention profiles showed on average 30%, 2%, 4% and 16% corresponding gene expression changes, respectively (Fig 3D). Overlap between histone modifications and gene expression was stronger for H3K4me3 and H3K27ac changes at promoters (8–81% overlap; S3A–S3C Fig) than for H3K27ac changes at enhancers (5–29% overlap; S3D Fig), indicating that enhancer modifications associate with direct gene expression to a lesser extent than promoter modifications.

To find out whether histone modifications at promoters and enhancers cooperate to regulate gene expression, we analyzed pairwise overlaps of genes associated with promoter- and enhancer classes (Fig 3E). The strongest overlaps occurred in classes belonging to the same profiles: P1a/P2a/E1 (Suppression), P2b/E2 (Down), P2c/E3 (Up) and P1b/P2d/E4 (Prevention) (Fig 3E). We found that 57% of the genes overlapping in the Suppression profile classes

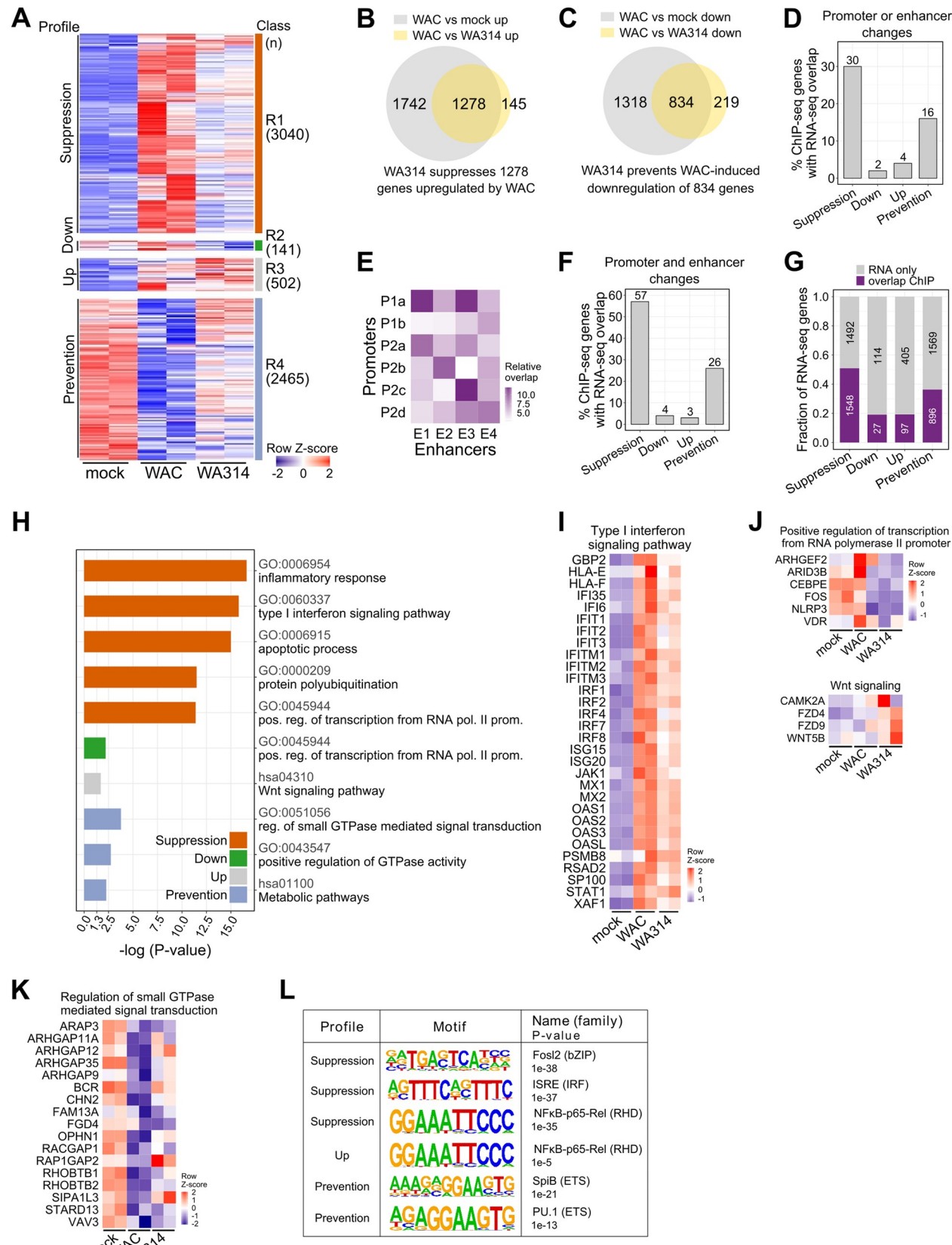

**Fig 3. Association between histone modifications and gene expression in *Y. enterocolitica* infected macrophages. A,** Heatmap from clustering of all DEGs from mock, WAC and WA314 comparisons. Clustering identified 4 major classes R1-R4 (color coded, right side). Profiles describing relation of expression levels between mock, WAC and WA314 are indicated on the left. In the heatmap, gene rlog counts are row-scaled (row Z-score). **B, C,** Venn diagrams of DEG overlaps between WAC vs mock up and WAC vs WA314 up **(B)** and WAC vs mock down and WAC vs WA314 down **(C)**. Overlaps show the number of genes whose up-regulation by WAC is suppressed by WA314 **(B)** and whose down-regulation by WAC is prevented by WA314 **(C)**. **D,** Barplot showing percentage of genes from ChIP-seq profiles with promoter or enhancer modifications overlapping with corresponding RNA-seq profiles from DEG analysis. **E,** Heatmap presentation showing relative overlap of genes in promoter classes P1a-P2d and enhancer classes E1-4. Light to dark color scale indicates low to high overlap. **F,** Barplot showing percentage of genes with coordinated promoter and enhancer changes from ChIP-seq profiles in **(E)** overlapping with corresponding RNA-seq profiles. **G,** Bar plot showing fraction and number (numbers in bars) of genes from RNA-seq profiles overlapping with genes from corresponding ChIP-seq profiles at promoters or enhancers. **H,** Bar plot showing enriched GO and KEGG (hsa prefix) terms for genes overlapping in RNA-seq and ChIP-seq as in **(G)**. Bars represent log10 transformed P-values correlating with significance of enrichment. Pos: positive, reg: regulation, pol: polymerase, prom: promoter. **I-K,** Heatmaps of row-scaled (row Z-score) RNA-seq rlog gene counts for genes with RNA-seq and ChIP-seq promoter or enhancer overlaps from pathways in **(H)**. **L,** Representative enriched transcription factor motifs in promoter and enhancer regions of genes with RNA-seq and ChIP-seq (promoter or enhancer) overlaps as in **(G)**.

and 26% of the genes overlapping in the Prevention profile classes showed corresponding changes in gene expression (Fig 3F). In comparison, only 4% and 3% of genes associated with histone modifications in the Down and Up profile classes, respectively, showed expression changes (Fig 3F). We conclude that reprogramming of histone modifications upon *Y. enterocolitica* infection in macrophages can cause significant changes in gene expression. The size of the effect displays a very wide range from 2 to over 80% and strongly depends on the localization (promoter or enhancers), cooperation (e.g. at promoters or between related promoters and enhancers) and profile of the histone modifications.

To address the question which biological pathways are regulated by the epigenetic reprogramming upon *Yersinia* infection, we subjected all DEGs in the different expression classes/profiles that are associated with histone modifications (Fig 3G) to Gene Ontology (GO) and KEGG analysis (Fig 3H). In profiles Suppression, Down, Up and Prevention, 51%, 19%, 19% and 36% of DEGs, respectively, were associated with at least one corresponding histone modification change (Fig 3G). Thus, the proportion of gene expression changes that is associated with histone modifications is clearly higher than the proportion of histone modifications that is associated with corresponding gene expression changes (compare Fig 3D and 3G). Suppression profile genes were most highly enriched in inflammatory response, type I interferon signaling and apoptotic process (Fig 3H) and mainly encode pro-inflammatory cytokines, chemokines, feedback regulators (like TNFAIPs) and type I IFN signaling mediators (Figs 3I and S4A). Down profile was enriched in regulation of transcription (Fig 3H) containing CEBPE and FOS (Fig 3J). CEBPE regulates myeloid lineage differentiation [68], inflammasome activation and interferon signaling [69]. FOS encoding c-Fos protein is a part of AP1 complex, which regulates macrophage differentiation and long range enhancer interactions [70], suppresses pro-inflammatory cytokine and IL-10 expression and increases lysosome mediated bacterial killing [71]. Up profile genes were enriched in Wnt signaling (Fig 3H and 3J), which directs macrophage differentiation, regulates expression of key anti-inflammatory mediators and modulates phagocytosis [72]. Prevention profile genes were enriched in pathways involving regulation of GTPases, including mostly Rho GTPase pathway genes (Fig 3H and 3K), and in metabolic pathways (Fig 3H), including 22 genes for the metabolism of cholesterol, fatty acids, acyl-CoA or glucose (S7 Table). Suppression profile genes associated with latent enhancers (Fig 2F) were enriched among others in negative regulation of transcription and inflammatory signaling (S4B Fig).

Pathogenic *Yersinia* exploit a variety of host signaling pathways for pathogenesis such as MAP kinase [28], PI3K/ Akt [73], cell death [62] and trafficking pathways, e.g., involving Rab GTPases [74]. Notably, gene expressions and histone modifications belonging to these pathways are also modulated by *Yersinia* T3SS effectors (S16 Table). For instance, *Yersinia* was

shown to inhibit activation of the MAPK pathway [28] and several MAPK kinases and phosphatases were found to be suppressed by plasmid-encoded virulence factors in the Suppression profile (S4C Fig). Furthermore, both Suppression and Prevention profiles contained genes from Rab GTPase signaling (S4D Fig), which is exploited by pathogenic *Yersinia* to promote intracellular survival [74].

Analysis of transcription factor (TF) motif enrichment revealed binding sites of inflammatory regulators from the RHD- (NFkB-p65-Rel), IRF- (ISRE, IRF2) and bZIP- (Fra2, Fosl2, Jun-AP1) families in Suppression profile genes (Figs 3L and S4E). No transcription factor motifs were enriched in Down profile. Up profile was enriched for RHD (NFkB-p65-Rel) binding sites (Figs 3L and S4E) indicating that *Yersinia* effector activities can lead to increased gene expression also through regulation of NF-κB signaling. Prevention profile genes were enriched for distinct motifs from ETS (SpiB, PU.1) family (Figs 3L and S4E). ETS TFs are known to interact extensively with other TFs [75] and play a role in cytokine gene expression [76–79] and macrophage differentiation [39].

Taken together, the histone modifications that are reprogrammed upon *Y. enterocolitica* infection associate with central transcriptional programs in macrophages. Most significantly, PAMP-induced up-regulation of immune signaling and inflammatory response genes is suppressed and the down-regulation of metabolic and Rho GTPase pathway genes is prevented by the bacteria.

## *Y. enterocolitica* reprogramms histone marks of Rho GTPase pathway genes

Interestingly, the Rho GTPase pathway genes were most significantly enriched in Prevention profile genes connected to histone modifications (Fig 3H and 3K). It was also found enriched in the Suppression profile and in the latent enhancer associated genes (S8 Table). Altogether, the dynamic histone modifications identified here were associated with 324 unique Rho GTPase pathway genes (61% of all known Rho GTPase genes; Fig 4A and S9 Table).

A great variety of bacterial virulence factors target or imitate Rho proteins or their regulators to manipulate immune cell functions [80,81]. Subversion of Rho GTPase activities is also a central virulence strategy of *Yersinia* and there is mediated by the T3SS effectors YopE, YopT and YopO/YpkA [25]. However, only scarce information is available on the epigenetic regulation of Rho GTPase pathway genes in general and relevant systems-level studies are lacking [82]. Considering the strong association of dynamic histone modifications with Rho GTPase pathway genes in *Yersinia* infected macrophages, we examined this connection in more detail here.

The Rho GTPase family is composed of 20 proteins, which can be divided into 8 subfamilies [83]. Rho GTPases are best known for regulating the actin cytoskeleton but also for control of vesicle transport, microtubule dynamics, cell cycle and gene expression [84,85]. Through these basic activities, they play central roles in immune cell functions such as phagocytosis, chemotaxis and adhesion [80]. The classical Rho GTPases cycle between inactive GDP-bound and active GTP-bound states (Fig 4H). Guanine nucleotide exchange factors (GEFs) promote the dissociation of GDP leading to GTP loading and activation [86]. Active GTP-bound Rho proteins stimulate effector proteins which carry out basic molecular activities [87,88]. GTPase activating proteins (GAPs) bind to GTP-loaded Rho proteins and thereby stimulate their intrinsic GTPase activity and lead to their deactivation [89]. Most GEFs, GAPs and effectors are not specific for individual Rho GTPases but associate with members of one or more subfamilies [87,90]. Currently 77 RhoGEFs [86], 66 Rho GAPs [89] and up to 370 Rho GTPase effectors are known, whereby the number of effectors may still be growing [87,91,92].

Notably, 68%, 62%, 58% and 74% of all known genes for GAPs, GEFs, effectors and Rho proteins, respectively, exhibited altered histone modification patterns in *Y. enterocolitica*

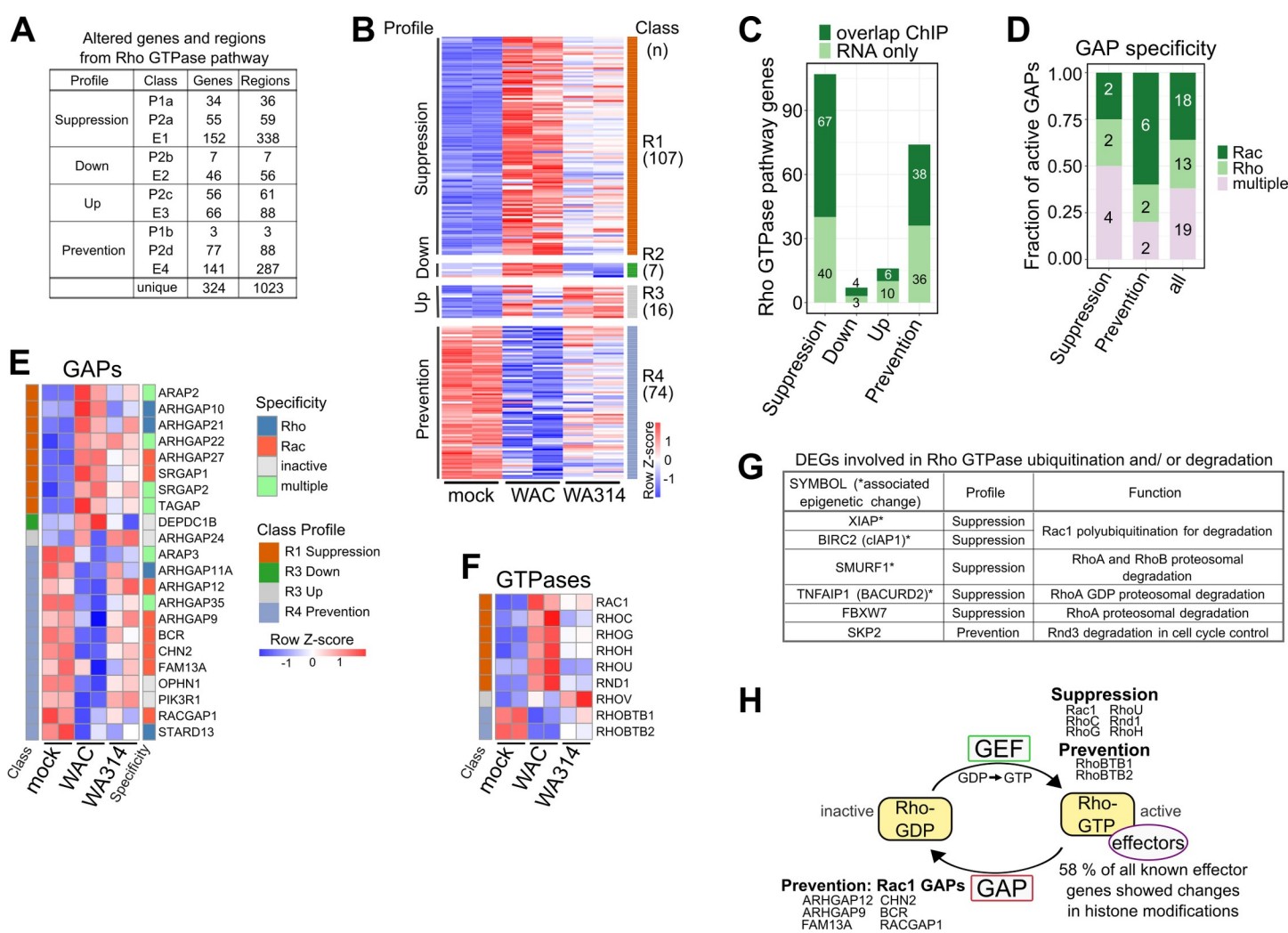

**Fig 4. Epigenetic and transcriptional regulation of Rho GTPase pathway genes in *Y. enterocolitica* infected macrophages. A,** Number of Rho GTPase pathway genes associated with changed regions in promoter (P1-b, P2a-d) or enhancer (E1-4) classes. "Unique" refers to total unique genes or regions (overlapping regions were merged together) when counting together all promoter and enhancer classes. **B,** Heatmap of all DEGs from Rho GTPase pathway belonging to classes R1-R4 (color coded, right side) in Fig 3A. Association with profiles is shown on the left side. "n" refers to the number of genes. Gene rlog counts were row-scaled (row Z-score). **C,** Bar plot showing number of DEGs belonging to Rho GTPase pathway from RNA-seq profiles in (**B**) overlapping with genes with corresponding promoter (P1a-b, P2a-d classes) and/ or enhancer (E1-4 classes) changes from the same profiles. **D,** Bar plot showing Rho GTPase specificity of active GAPs [90] for genes with RNA-seq and ChIP-seq overlaps in (**C**). Fraction and number (in bars) for all active GAPs and for GAPs in the Suppression and Prevention profiles are shown. **E, F,** Heatmaps of RNA-seq rlog counts for GAP (**E**) and Rho GTPase genes (**F**) with RNA-seq and ChIP-seq overlaps in (**C**). Associated classes are indicated on the left and specificity of GAPs for Rho GTPases is indicated on the right. Gene rlog counts were row-scaled (row Z-score). **G,** Table showing DEGs (from Fig 3A and S6 Table) known to regulate ubiquitination and/ or proteosomal degradation of Rho GTPases. **H,** Schematic representation of Rho GTPase cycle showing reprogrammed Rho GTPase pathway genes divided into GAPs, GEFs, Rho GTPases and effectors, and assigned to profiles.

infected macrophages (S5A Fig "RNA & ChIP" and "ChIP only" fractions). Histone modifications were concomitantly altered at enhancers and promoters of 127 Rho GTPase pathway genes (S5B Fig). More than 2 enhancer regions in the E1 and E4 enhancer classes were on average associated with an individual Rho GTPase pathway gene (Fig 4A) and the number of enhancers per Rho GTPase gene ranged from 1 to 14 (S5C Fig and S9 Table). FNBP1, RAP-GEF1 and ELMO1 were the genes for which 13, 13 and 14 putative enhancers, respectively, were found (S9 Table).

To draw the most informed conclusion about the biological significance of Rho GTPase pathway gene regulation by *Yersinia*, we created a heatmap of all Rho GTPase pathway DEGs

in mock-, WAC-, and WA314-infected macrophages independent of their association with histone modifications [87,90–94]. This revealed altogether 204 DEGs in Suppression-, Down-, Up- and Prevention profiles (classes R1-R4, Fig 4B and S10 Table). Of these, 29 encode GTPase activating proteins (GAPs), 31 guanine nucleotide exchange factors (GEFs), 134 effectors and 11 Rho GTPases (S10 Table). 115 (56%) of these DEGs were in fact associated with dynamic histone modifications (Fig 4C).

Recently all known proteins with putative GAP or GEF activities were biochemically characterized regarding their actual effect on the three main Rho family proteins RhoA, Rac1 and Cdc42 [90]. Analysis of our data in view of this report revealed that in the Prevention profile 60% of the active GAPs (6/10) act specifically on Rac as compared to 36% (18/50) of all known GAPs (Fig 4D and 4E and S11 Table) [90]. In the Suppression profile, Rho-, Rac- or Cdc42 specific GAP genes are not enriched and the percentages of GAP activities acting on Rho (25%) or Rac (25%) are identical to those of total GAPs (Fig 4D and 4E) [90]. We conclude that aided by epigenetic remodelling *Y. enterocolitica* tends to keep Rac-inhibiting GAPs at the level of uninfected cells.

Of the 8 Rho GTPase genes in the Suppression and Prevention profiles, three encode classical Rho proteins (Rac1, RhoC, RhoG) and five encode atypical Rho proteins (RhoH, RhoU, RND1, RhoBTB1, RhoBTB2) (Fig 4F). Most atypical Rho proteins are considered constitutively active and consequently are not regulated by GEFs and GAPs but instead are controlled at the level of transcription and targeted destruction, i.e. proteasomal degradation [95,96]. The cellular levels of RhoC and RhoG were also shown to be transcriptionally regulated [97,98]. Classical and atypical Rho proteins often have overlapping cellular functions and share the same effector proteins. Atypical Rho proteins have so far been implicated in multiple activities including tumour suppression (e.g. RhoBTB2), cell transformation and -morphogenesis as well as development (e.g. RhoU and Rnd) [85,95,99,100]. Many atypical Rho proteins have been found to regulate the activity of classical Rho Proteins, e.g. RhoH spatially controls Rac1 activity [101] and Rnd proteins bind and steer Rho GAP proteins to specific cell sites [99]. Notably, the Rho protein genes in the Suppression profile include Rac1, the Rac1 activator RhoG and the atypical Rho proteins RhoH and RhoU (Fig 4F). These Rho proteins can alternatively activate Rac effectors, spatiotemporally control Rac activity or take over Rac functions in cells [85,87,95,100,101]. Thus, a prominent effect of *Yersinia* in the Suppression profile is to inhibit the expression of genes, whose products can keep up Rac activity. This effect complements well the blocked downregulation of Rac GAPs in the Prevention profile (Fig 4D and 4E). Altogether epigenetically controlled gene expression might cause low Rac activity in *Yersinia* infected cells and thereby cooperate with the Rac down-regulating T3SS effectors YopE, YopO and YopT [25].

66 Rho GTPase effectors were altogether found in the overlaps of gene expression and epigenetic profiles (S5A, S5D and S5F Fig) [87,91,92]. This includes effectors for all eight Rho GTPase subfamilies (Rho, Rac, Cdc42, RhoD/Rif, Rnd, Wrch-1/Chp, RhoH and RhoBTB) (S5F Fig and S12 Table). Interestingly, while *Yersinia* prevents downregulation of genes for the atypical RhoBTBs (Fig 4F), it at the same time suppresses expression of genes encoding specific RhoBTB effectors (S5F Fig and S12 Table). Furthermore, 19 of the identified Rho GTPase effectors have been implicated in epigenetic and transcriptional regulation, suggesting mechanisms for crosstalk and feedback in the epigenetic control of Rho GTPase pathway gene expression (S12 Table). Another level of regulation is provided by the finding that some DEGs from Suppression- and Prevention profile genes belong to the machinery for ubiquitination and/or proteasomal degradation of Rho proteins (Fig 4G) [85].

Altogether these data indicate that yersiniae modulate expression of a large part of Rho GTPase pathway genes in macrophages through reprogramming of cellular chromatin on multiple levels (Figs 4E, 4F and 4H and S5D and S5E).

## YopP is a major modulator of epigenetic reprogramming upon *Y. enterocolitica* infection

The *Y. enterocolitica* T3SS effectors YopP (YopJ in *Y. pseudotuberculosis*) and YopM have been shown to modulate inflammatory gene expression in *Yersinia* infected macrophages [29]. We therefore examined whether YopP and YopM contribute to the epigenetic reprogramming by *Y. enterocolitica*. Macrophages were infected (6 h) with WA314 strains lacking YopP or YopM (strains WA314ΔYopP or WA314ΔYopM; S1 Table) and investigated using ChIP-seq and RNA-seq (H3K4me3 and H3K27ac modifications; Fig 1A). Principal component analysis (PCA) revealed that the WA314ΔYopM- and WA314 induced H3K27ac- and H3K4me3 modifications located close to each other, while the WA314ΔYopP induced modifications were clearly separate (Figs 5A and S6A). This suggests that YopP but not YopM significantly contributes to the epigenetic changes produced by WA314. The effects of WA314ΔYopP and WA314ΔYopM were directly compared with the effects of WA314 on H3K27ac- and H3K4me3 modifications and gene expression. Notably, 684 H3K4me3 regions and 5094 H3K27ac regions were differentially regulated between WA314ΔYopP and WA314 (Fig 5B). In contrast, altogether only 29 histone modifications were differentially regulated between WA314ΔYopM and WA314 (Fig 5B). While 1616 DEGs were detected between WA314ΔYopP and WA314, also 804 DEGs were found between WA314ΔYopM and WA314 (Fig 5B). This suggests that YopP affects gene transcription in macrophages to a large extent through modulation of H3K27ac- and H3K4me3 histone modifications. In contrast, although YopM strongly affects gene transcription, it does not do so by regulating H3K4me3- and H3K27ac histone modifications. We next studied what proportions of the WA314-induced histone modifications in the Suppression-, Prevention-, and Up- and Down profiles were due to YopP (S6B Fig). For this, the percentage YopP effect was calculated from the ratio of fold change (FC) between WA314ΔYopP vs WA314 and either WA314 vs WAC (Suppression and Prevention profiles) or WA314 vs mock (Up and Down profiles). In the promoter and enhancer classes the median Yop contribution to the WA314 effects was 42% and ranged from 8.9% - 57.2% (S6B Fig). Furthermore, the median YopP contribution to the WA314 effects on gene expression associated with histone modifications was 51.4% (S6C Fig). We conclude that YopP on average contributes around one half to the effects of *Y. enterocolitica* on histone modifications and gene expression. Consequently, other T3SS associated virulence factors—except YopM– also contribute significantly.

We looked in more detail at the effects of YopP on Rho GTPase pathway genes (Fig 4 and S13 Table). In the Suppression- and Prevention profiles of Rho GTPase pathway genes, YopP contributed in the mean 48.5% and 78.5%, respectively, to the WA314 effects on gene expression (S13 Table). However, at the level of individual genes the YopP effect was widely spread from 2 to 103% in the Suppression- and from 29 to 177% in Prevention profile (Fig 5C and S13 Table). We tested whether this widely distributed effect of YopP also applies to inflammatory genes (Figs 3H and S4A and S14 Table). Notably, while the mean effect of YopP on the Suppression profile inflammatory genes was 46% (S14 Table), the effect on individual genes was spread from -57 to 103% (Fig 5D). Although the majority of YopP induced changes in gene expression were associated with corresponding YopP induced changes in histone modifications, some YopP regulated genes were associated with histone modifications induced by other T3SS effectors and vice versa (e.g. IL2RA highlighted in Fig 5D and 5E; S13 and

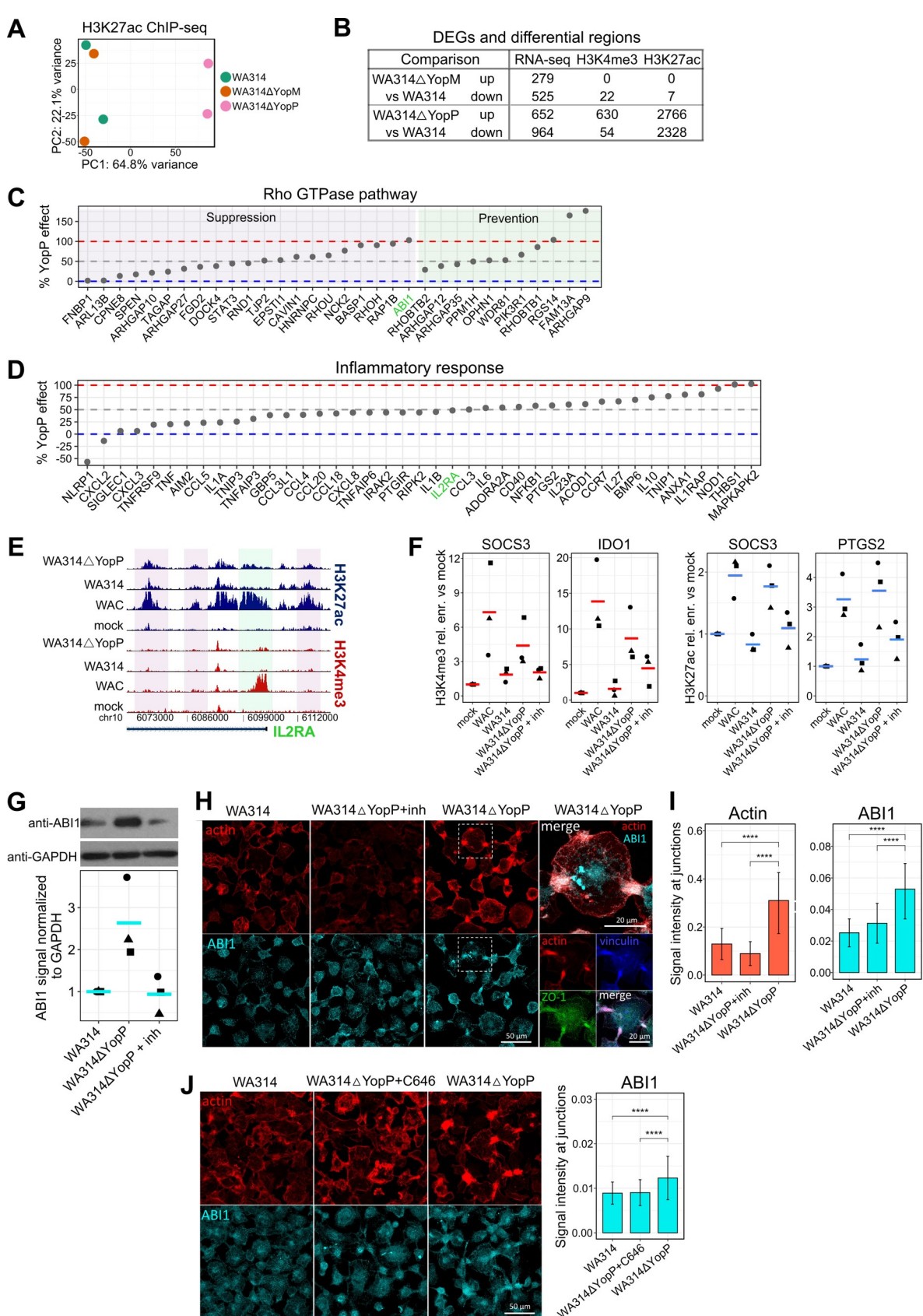

**Fig 5. Role of YopP and YopM in the epigenetic and transcriptional reprogramming of macrophages by *Y. enterocolitica*. A,** Principal component analysis of H3K27ac tag counts in classes P1a-P2d and E1-4 in two replicates of macrophages infected with the indicated strains. **B,** Number of statistically significant ($\geq$ 2-fold change, adjusted P-value $\leq$ 0.05) DEGs and differentially enriched H3K4me3 and H3K27ac regions for WA314ΔYopM vs WA314 and WA314ΔYopP vs WA314. **C, D,** Percentage of WA314 effect on expression of Rho GTPase pathway **(C)** or Inflammatory response **(D)** genes that is caused by YopP. Genes with associated promoter or enhancer changes from the same profile and significant change for RNA-seq (DESeq2) and ChIP-seq (diffReps) between WAC vs WA314 are shown. The percentage value was calculated from the ratio of fold change (FC) between WA314ΔYopP vs WA314 and WA314 vs WAC for Suppression **(C, D)** and Prevention **(C)** profiles. **E,** Peak tracks of ChIP-seq tag densities showing H3K4me3 and H3K27ac changes associated with IL2RA gene, whose suppression of gene expression by WA314 is produced in a YopP-dependent manner **(D)**. H3K4me3 and H3K27ac densities at the IL2RA promoter (green shade) and H3K27ac densities at the enhancers (purple shade) are not affected by YopP (compare effects of WA314 and WA314ΔYopP). **F,** Dot plots of H3K4me3 and H3K27ac ChIP-qPCR signals at indicated genes in macrophages not infected (mock) or infected with indicated strains. WA314ΔYopP infections occurred in the absence or presence of MAPK inhibitors (inh). The ChIP-qPCR signal was expressed as relative (rel.) enrichment (enr.) vs mock. Lines represent mean of 3 biological replicates (dots with different shapes). SOCS3: Suppressor of cytokine signaling 3, IDO1: Indoleamine 2,3-Dioxygenase 1, PTGS2: Prostaglandin-Endoperoxide Synthase 2. **G,** Western blot analysis (top) and quantification (bottom) of ABI1 levels in primary human macrophages infected with WA314, WA314ΔYopP and WA314ΔYopP+inh (MAPK inhibitors) at an MOI of 100 for 6 h. GAPDH was used as a loading control and for normalization of ABI1 signal. Western blot is representative of three independent experiments shown in quantification. **H,** Immunofluorescence staining of primary human macrophages infected with WA314, WA314ΔYopP+inh (MAPK inhibitors) or WA314ΔYopP with MOI of 100 for 6 h. Cells were stained with Alexa568 phalloidin (red) to visualize actin and antibodies for ABI1 (cyan), vinculin (blue) and ZO-1 (green). Fourth column shows an overlay of actin and ABI1 (top) and actin, vinculin and ZO-1 (bottom) for WA314ΔYopP infection. Data are representative of two independent experiments. **I,** Quantification of actin (left) and ABI1 (right) signal intensity at cell junctions from immunofluorescence of primary human macrophages infected with WA314, WA314ΔYopP+inh (MAPK inhibitors) or WA314 YopP with MOI of 100 for 6 h. Data are representative of two independent experiments. **J,** Immunofluorescence staining of primary human macrophages infected with WA314, WA314ΔYopP+C646 (p300 inhibitor) or WA314ΔYopP with MOI of 100 for 6 h (left). Cells were stained with Alexa568 phalloidin (red) to visualize actin and antibody for ABI1 (cyan). Quantification of ABI1 signal at cell cell contacts is shown on the right. Data are representative of two independent experiments. In **(I)** and **(J)** bars show mean and error bars represent standard deviation. ****: P-adjusted < 0.0001 by unpaired Wilcoxon test.

S14 Tables and S6D Fig). Further, although YopP contribution to gene expression and histone modifications was evident for the majority of genes and regions associated with inflammatory response and Rho GTPase pathways, the YopP effects on gene expression on one hand and histone modifications on the other hand frequently did not correlate (S6D Fig). Thus, the YopP effect on the expression of individual Rho GTPase pathway and inflammatory genes varies considerably from representing the complete *Yersinia* wild type effect to just a minor contribution. It thereby appears that different T3SS effectors and histone modifications contribute to the widely varying YopP effects on gene expression.

YopP/YopJ blocks PAMP-induced inflammatory gene expression by inhibiting NF-κB and MAP-kinase signaling [29]. Interestingly, MAP-kinases also phosphorylate histone-3 at serine-10, a modification that is thought to promote the deposition of activating histone marks like H3K14ac and H3K16ac [102]. We therefore sought to find out whether YopP reduces the deposition of H3K4me3- and/or H3K27ac marks through inhibition of MAP-kinases. WA314 inhibited the deposition of H3K4me3 at the IDO1- and SOCS3 promoters and deposition of H3K27ac at the SOCS3 and PTGS2 promoters in the macrophages (Fig 5F). YopP contributed to the WA314 effects on these genes, as seen by significantly less inhibitory activity of WA314ΔYopP on deposition of the histone marks when compared to WA314 (Fig 5F). When combined with the MAP-kinase inhibitors SB203580 and PD98059, targeting p38 and MEK1, respectively, WA314ΔYopP was nearly as effective as WA314 in inhibiting the deposition of H3K4me3 marks at the IDO1 and SOCS3 promoters and deposition of H3K27ac marks at the PTGS2 and SOCS3 promoters (Fig 5F). Thus, MAP-kinase inhibitors can substitute for the missing YopP activity in WA314ΔYopP. This is consistent with the idea that YopP blocks deposition of the histone marks through MAP-kinase inhibition.

Finally, we wanted to determine whether the histone modification-driven reprogramming of Rho GTPase pathway genes has functional consequences in macrophages. For this, we took advantage of suppression of ABI1 gene expression by WA314, which is entirely dependent on YopP (Figs 5C and S6D). ABI1 encodes a component of the WAVE regulatory complex that

controls actin polymerization in macrophages [103,104]. ChIP-qPCR analysis confirmed that YopP suppresses H3K27ac at enhancer within ABI1 gene and H3K4me3 at ABI1 promoter (S6E Fig). Interestingly, during WA314ΔYopP infection treatment with MAP kinase inhibitors suppressed only H3K4me3 and not H3K27ac (S6E Fig), suggesting that different signaling pathways regulate promoter H3K4me3 and enhancer H3K27ac. Gene expression analysis by RT-qPCR confirmed suppression of ABI1 expression by YopP, which was dependent on inhibition of the MAPK pathway (S6F Fig). Corresponding to gene expression, the ABI1 protein was upregulated in WA314ΔYopP- compared to WA314 infected macrophages and this could be prevented by MAP-kinase inhibitors (Fig 5G). In the WA314ΔYopP infected macrophages ABI1 prominently accumulated at cell-cell contacts, where it colocalized with actin, vinculin and the zonula occludens protein ZO1 (Fig 5H and 5I). Further, accumulation of ABI1 was associated with increased levels of actin at the cell, cell contacts, indicating enhanced WAVE regulatory complex-induced actin polymerization (Fig 5H and 5I) [105]. Notably, accumulation of ABI1 and actin was reversed by treatment of p300 histone acetyltransferase inhibitor C646 (Figs 5J and S6G). Altogether this data shows that through inhibition of the MAPK pathway YopP targets histone modifications and gene expression of the ABI1 gene, causing a redistribution of ABI1 and actin in *Yersinia* infected macrophages.

## Discussion

At present only scarce information is available on epigenetic reprogramming of immune cells by pathogenic bacteria [48,106]. This study was conducted to find out whether the enteropathogen *Y. enterocolitica* alters chromatin states to globally control gene expression in human macrophages. Our results provide a number of novel insights into systemic reorganization of epigenetic histone modifications in human macrophages upon infection of pathogenic yersiniae.

A primary goal was to map in detail whether and how the effects of the *Yersinia* PAMPs on histone modifications are reorganized by the T3SS-associated virulence factors of the bacteria. To this end, we systematically compared the effects of a *Y. enterocolitica* wild type strain and a derived avirulent strain. Infection with virulent and avirulent yersiniae extensively reorganized histone marks (H3K4me1, H3K4me3 and H3K27ac) at macrophage gene promoters and enhancers. H3K27ac marks were the most dynamic ($> 22.000$ loci altered) followed by H3K4me1 marks (2156 loci altered). That H3K27ac marks are highly dynamic in human macrophages stimulated with bacterial agents has been reported previously [64].

We could classify the bacteria-induced reorganization of histone modifications into four profiles, which we named Suppression, Down, Up and Prevention. In the Suppression- and Prevention profiles, the bacteria's T3SS effectors block the PAMP-induced deposition or removal, respectively, of histone marks. In Down- and Up profiles, the T3SS-associated effectors down- or up-regulate, respectively, histone marks independently of PAMP activities.

The fraction of histone modifications overlapping with actual gene expression in the four profiles varied widely. Overlaps ranged from 30% in the Suppression profile to 2% in the Down profile. The biological significance of *Yersinia*-modulated histone modifications that do not immediately affect gene expression is unclear. It has been observed that chromatin changes in stimulated macrophages do not always associate with gene expression [10,12,107]. These "silent" histone modifications may be involved in priming, immune memory, immune tolerance or other complex gene regulatory processes in macrophages [38,39].

It was interesting to note that different histone marks at individual promoters or at related promoters and enhancers were reorganized in a coordinated manner upon infection, mostly

in the Suppression- and Prevention profiles. In these instances, the histone modifications were more frequently associated with changes in gene expression. Thus, the activity of bacterial pathogenicity factors lead to systematic co-regulation of different histone modifications to effectively act on gene transcription.

The T3SS effector YopP accounted for 40–50% of the changes in histone modifications and gene expression produced by virulent *Y. enterocolitica* in macrophages, as exemplified with inflammatory and Rho GTPase pathway genes. Inhibition of MAP-kinase signaling was a crucial mechanism underlying these YopP effects for selected target genes. YopP regulated the expression of genes, at which it changed histone modifications, to very different degrees. E.g., for individual inflammatory or Rho GTPase pathway genes the contribution of YopP to the *Yersinia* effect ranged from null to 100%. Interestingly, the histone modifications at some of the genes whose expression was strongly affected by YopP were changed by other T3SS effectors. We could exclude YopM as one of these effectors, because it did not alter close to any of the histone modifications investigated here. Thus, YopP but not YopM regulates the expression of selective genes through histone modifications and thereby seemingly cooperates to varying extents with other T3SS effectors. Of note, in addition to histone modifications, other gene regulatory mechanisms clearly also contribute to the *Yersinia*- and YopP effects on gene expression [108]. Thus, our data suggest that virulent yersiniae fine tune expression of individual genes in specific biological pathways through governing the cooperation between histone modifications and other gene regulatory mechanisms. For this the bacteria employ YopP as the main T3SS effector that cooperates with other effectors. Together the effectors mostly remodel PAMP-induced signal transduction but also exert PAMP independent effects on gene transcription [108].

Among the most significant findings of our study was the remodeling of histone modifications at 61% of all 534 known Rho GTPase genes. When considered in terms of gene expression, 38% of all known Rho GTPase pathway genes were differentially expressed whereby well over half of them were associated with dynamic histone modifications. The latter encode Rho proteins (Rho GTPases) and their effectors, activators (GEFs) and deactivators (GAPs). To control dynamic cellular processes such as cytoskeletal reorganization or vesicular transport, the right Rho proteins, regulators and effectors have to be activated or deactivated at the right time and location. Rho GTPase signaling networks are characterized by the sheer countless interactions that can take place between the different Rho GTPases, regulators and effectors and by crosstalk and feedback loops [87,90–92]. Our results add a further level of complexity to Rho GTPase regulation in macrophages by revealing an extensive epigenetic control of Rho GTPase pathway gene expression.

One of our intriguing findings was that genes encoding GAPs for Rac were overrepresented in the Prevention profile. This indicates that epigenetic changes upon *Yersinia* infection create a cellular state in which downregulation of Rac inhibitors is prevented, which favours Rac inhibition. Inhibition of Rac is a known central strategy of *Yersinia* with the three T3SS effectors YopE, YopT and YopO acting as Rac inhibitors [25]. Further consistent with this bacterial strategy is the finding that expression of Rac1, the Rac1 activator RhoG and the atypical Rho proteins RhoH and RhoU, which can take over Rac functions [85,87,95,100,101], were suppressed by the bacteria. A reason why *Yersinia* places a focus on inhibition of Rac may be the outstanding role that it plays in anti-microbial activities of macrophages such as phagocytosis, chemotaxis and production of reactive oxygen species [109]. Rac inhibition by YopE also prevents overshooting translocation of T3SS effectors into cells [25]. A number of genes for atypical Rho proteins (RHOH, RHOU, RND1, RHOBTB1, RHOBTB2) were found in the Suppression and Prevention profiles. The activities of atypical Rho proteins are generally controlled by gene transcription and protein degradation and not by GAPs and GEFs [95,96], but

their regulation by epigenetic mechanisms has so far not been reported. Although the specific functions of atypical Rho proteins in macrophages are widely unknown, atypical Rho proteins also can regulate the activity of classical Rho proteins in different cell types[85,95,99,100]. For instance, RhoH spatially controls Rac1 activity [101] and Rnd proteins bind and steer Rho GAP proteins to specific cell sites [99]. We also found that downregulation of genes for the atypical Rho proteins RhoBTBs was prevented and expression of genes for some RhoBTB effectors was suppressed in parallel by the bacteria. Thus, through epigenetic mechanisms *Yersinia* may determine which atypical Rho protein/effector pairs are functional in macrophages. Considering the universal role that Rho GTPases play in cellular function, the multifaceted epigenetic regulation of their expression will undoubtedly affect their activities and thereby have functional consequences in macrophages.

Providing support for this notion, we here show that MAP kinase (MAPK) pathway regulates histone modifications, gene expression and protein levels of ABI1 gene with associated redistribution of actin cytoskeleton and ABI1 protein, a component of the WAVE actin regulatory complex, at macrophage-macrophage contacts. Furthermore, treatment of macrophages with p300 histone acetyltransferase inhibitor reversed this ABI1 and actin accumulation, providing a direct link between histone modifications and actin reorganization. Increased ABI1 enhances actin accumulation and recruitment of adhesion proteins at these contacts, likely promoting the stabilization of these structures [110,111]. Taken together, in this study we decipher key principles of epigenetic reorganization of human macrophages upon infection of the bacterial enteropathogen *Y. enterocolitica*. 1) Remodeling of histone modifications and associated changes in gene expression can be classified in different profiles, with virulent bacteria mainly suppressing or preventing the effects of bacterial PAMPs, but also exerting independent stimulatory or inhibitory activities through their T3SS effectors. Suppressed histone modifications with corresponding gene suppression mostly belong to immune and inflammatory signaling, whereas prevented modifications/genes belong to Rho GTPase pathway and metabolic pathways (Fig 6). 2) Changes in the different histone modifications at promoters and enhancers are often coordinated, so that they can act cooperatively on gene expression. 3) The T3SS effector YopP contributes up to 50% to *Yersinia*-induced remodeling of histone modifications and gene expression, through its inhibitory effect on MAP kinase signaling. At the level of individual gene expression, the contribution of YopP varies from null to 100%, even in genes belonging to the same biological pathways. 4) Histone modifications, transcription factor expression, and other unidentified mechanisms are reprogrammed by YopP and other T3SS effectors and cooperatively cause the effects of *Yersinia* on individual gene expression. 5) While there was an up to 30% overlap between histone modifications and associated gene expression in the profiles, the majority of histone modifications did not alter gene expression. In these cases, such epigenetic modifications could prepare cells for subsequent stimuli or provide a basis for innate immune memory or tolerance. Further studies are warranted to test these intriguing possibilities.

## Materials and methods

### Ethics statement

Approval for the analysis of anonymized blood donations (WF-015/12) was obtained by the Ethical Committee of the Ärztekammer Hamburg (Germany).

### Bacterial strains

*Yersinia enterocolitica* strains used in this study are derivatives of the serotype O:8 strain WA314 harboring the virulence plasmid pYVO8 [112]. WA314ΔYopM was constructed by

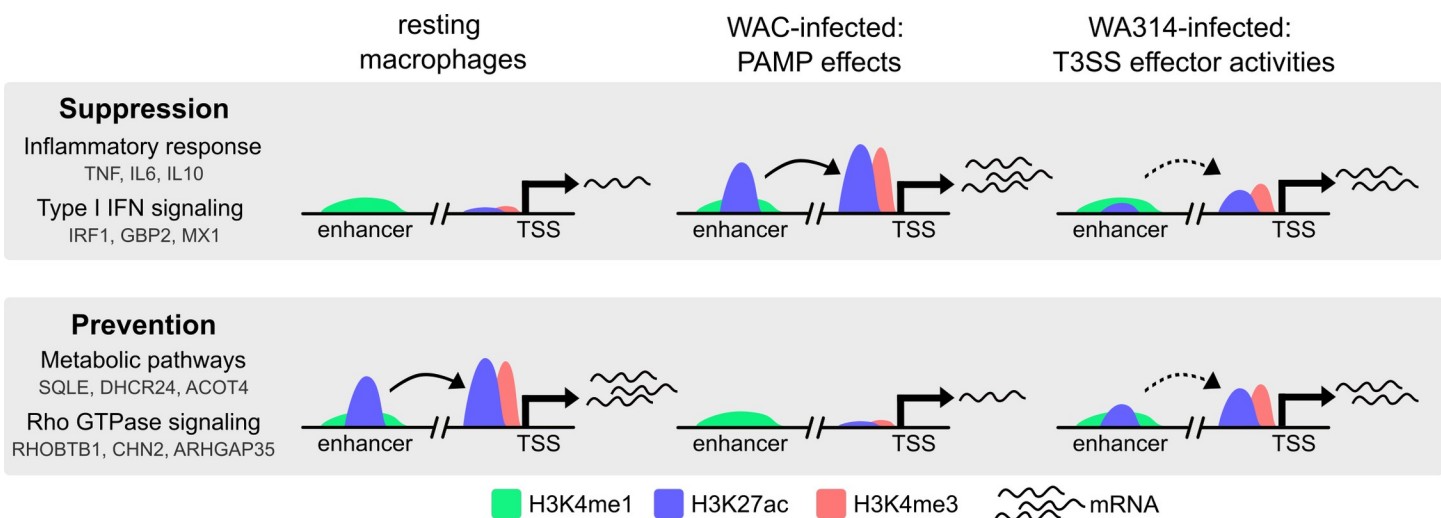

**Fig 6. Summary schematic showing the main effects of *Y. enterocolitica* on macrophage chromatin in gene expression.** Plasmid-encoded virulence factors suppress upregulation (Suppression profile) and prevent downregulation (Prevention profile) of PAMP induced H3K27ac and H3K4me3 changes at gene promoters and/ or enhancers and consequently the expression of associated genes. Suppression profile is associated with genes from Inflammatory response and Type I IFN signaling, whereas Prevention profile is associated with genes from Metabolic pathways and Rho GTPase signaling. Representative genes belonging to each pathway are shown. Enhancer activity is indicated with arched arrows. Increased promoter activity is indicated with more mRNAs.

replacing the YopM gene in WA314 with a kanamycin resistance cassette [113]. WA314ΔYopP was generated by insertional inactivation of *yopP* gene [114]. WA314-YopE-ALFA strain was constructed using CRISPR-Cas12a-assisted recombineering approach as described in preprint from Carsten et al., 2021 [115]. S1 Table provides an overview of the bacterial strains used in this study and oligonucleotide sequences for the generation of WA314-YopE-ALFA strain.

## Cell culture

Human peripheral blood monocytes were isolated from buffy coats as described in Kopp et al., 2006 [116]. Cells were cultured in RPMI1640 containing 20% autologous serum at 37˚C and 5% $CO_2$ atmosphere. The medium was changed every three days until cells were differentiated into macrophages after 7 days. Macrophages were used for infection 1 week after the isolation except for RNA-seq experiments for two mock, WA314 and WA314ΔYopM samples from Berneking et al., 2016 [36] and WA314ΔYopP RNA-seq samples where cells were used after 2 weeks of differentiation (RNA-seq samples from batch 2 in S17 Table).

## Infection of cells

On the day before infection of primary human macrophages the cell medium was changed to RPMI1640 without antibiotics and serum and precultures of *Y. enterocolitica* strains (S1 Table) were grown overnight in LB medium with appropriate antibiotics at 27˚C and 200 x rpm. On the day of infection precultures were diluted 1:20 in fresh LB medium without antibiotics and incubated for 90 min at 37˚C and 200 x rpm to induce activation of the *Yersinia* T3SS machinery and Yop expression. Afterwards bacteria were pelleted by centrifugation for 10 min at 6000 x g, 4˚C and resuspended in 1 ml ice-cold PBS containing 1 mM $MgCl_2$ and $CaCl_2$. The optical density $OD_{600}$ was adjusted to 3.6 and afterwards macrophages were infected at multiplicity-of-infection (MOI) of 100. Cell culture dishes were centrifuged for 2 min at RT and 200 x g to sediment bacteria on the cells and synchronize infection. Cells were incubated at 37˚C for 6 h.

## CFU assay

For analysis of bacterial growth (S1E Fig) *Yersinia* cultures were prepared and cells infected as described above in "Infection of cells". Bacterial sample at 0 h was prepared from bacteria used as an input. Afterwards macrophages and bacteria present in the wells were harvested by scraping without washing and centrifuged at 5000 x g at 4˚C. Pellet was resuspended in 0.5% Digitonin in PBS to lyze macrophages and not bacteria and incubated for 10 min on ice. Afterwards lysate was used to prepare 10-fold serial dilutions, which were plated in duplicate on LB agar plates containing appropriate antibiotics. Plates were incubated at 26˚C for 24 h before counting the colonies. Two independent experiments with different biological replicates were used for the analysis. The results showed that the number of wild type *Yersinia* at 6 h increased about 3 fold when compared to 0 h (S1E Fig) indicating a low level of replication under selected infection conditions.

## SYTOX Green assay

For analysis of cell death during infection SYTOX Green assay (Thermo Fisher Scientific) was performed which measures membrane permeability [117]. $6x10^4$ macrophages per well in a 96-well plate were seeded in RPMI1640 on the day before analysis. The following day macrophages were treated with PBS (mock), WA314 (MOI 100) or positive control (1% TRITON-X-100) in duplicates. Plates were centrifuged for 2 min at RT, 200 x g and incubated at 37˚C for 6 h. SYTOX Green reagent was added to cells 30 min before the end of 6 h incubation. SYTOX fluorescence due to membrane permeabilization was measured with plate reader (Tecan infinite M200) at 485/520 nm. Two biological replicates were used for the analysis.

## Flow cytometry

Antibody staining for flow cytometry was done in FACS buffer (PBS, 0.1% BSA) at room temperature in the dark for 30 min, followed by washing in FACS buffer and fixation with 2% paraformaldehyde in FACS buffer. Cells were stained with the following antibodies: anti-CD14 (Biolegend, 301806), anti-CD16 (Biolegend, 302018), anti-CD38 (Biolegend, 301806), anti-CD86 (BD biosciences), anti-HLA-DR (Biolegend, 307642), anti-CD163 (Biolegend, 333608) and anti-CD206 (Biolegend, 321109). In addition, cells were stained with live/dead dye (Invitrogen, P30253) to identify live cells.

Flow cytometry data was acquired on a Celesta flow cytometer (BD biosciences) and flow cytometry data was analyzed using Flowing software (Turku Bioscience). Analysis showed that 63.9% of macrophage culture were CD14+ macrophages (S1F Fig). CD14+ macrophages were positive for macrophage markers CD38 (99%), CD86 (98.5%), HLA-DR (74.9%), CD206 (91.6%), CD163 (61.6%) and CD16 (88%) (S1F Fig).

## Treatment with inhibitors

For the MAPK pathway inhibition combination of 5 μM SB203580 (Cayman Chemical) and 5 μM PD98059 (Merck Millipore), which target p38 and MEK1, respectively, was used. Inhibitors were added to macrophages 30–60 min before the infection.

Staurosporine (LC Laboratories; 5 μM) or C646 (Merck; 10 μM) were added at the start of infection.

## RNA-seq

Total RNA of $1–2 \times 10^6$ human macrophages (2–4 biological replicates/ macrophage donors per condition, see S17 Table) was isolated using RNeasy extraction kit (Qiagen) including

DNAse treatment according to manufacturer's instructions. RNA integrity of the isolated RNA was analyzed with the RNA 6000 Nano Chip (Agilent Technologies) on an Agilent 2100 Bioanalyzer (Agilent Technologies). mRNA was extracted using the NEBNext Poly(A) mRNA Magnetic Isolation module (New England Biolabs) and RNA-seq libraries were generated using the NEBNext Ultra RNA Library Prep Kit for Illumina (New England Biolabs) as per manufacturer's recommendations. Concentrations of all samples were measured with a Qubit 2.0 Fluorometer (Thermo Fisher Scientific) and fragment lengths distribution of the final libraries was analyzed with the DNA High Sensitivity Chip (Agilent Technologies) on an Agilent 2100 Bioanalyzer (Agilent Technologies). All samples were normalized to 2 nM and pooled equimolar. The library pool was sequenced on the NextSeq500 (Illumina) with 1 x 75 bp (1 x 51 bp for WA314ΔYopP samples) and total 19.9 to 23.8 million reads per sample.

### RNA-seq analysis

Sequencing reads containing bases with low quality scores (quality Phred score cutoff 20) or adapters were trimmed using TrimGalore program (http://www.bioinformatics.babraham.ac.uk/projects/trim_galore/).

Reads were aligned to the human reference assembly hg19 using STAR [118]. Feature-Counts [119] was employed to obtain the number of reads mapping to each gene.

RNA-seq data have been deposited in the ArrayExpress database at EMBL-EBI (www.ebi.ac.uk/arrayexpress) under accession number E-MTAB-10473. RNA-seq data from additional replicates of mock, WA314 6h and WA314ΔYopM 6h were obtained from European Nucleotide Archive (ENA) at http://www.ebi.ac.uk/ena/data/view/PRJEB10086.

### RNA-seq differential expression analysis

Statistical analysis of differential expression was carried out with DESeq2 [120] using raw counts as an input and the experimental design *~batch + condition*. Significantly enriched genes were defined with fold change ≥ 2 and adjusted P-value ≤ 0.05. Normalized rlog counts for each gene were obtained after batch effect removal with limma package [121] and used for downstream clustering analysis and visualization. PCA analysis and sample distance heatmaps confirmed high reproducibility between all biological replicates (see S17 Table) used in the analysis.

Clustering analysis of all DEGs from comparisons between mock, WAC, WA314 was done with rlog counts in R with pheatmap package. Clustering was performed with clustering distance based on Euclidean distance and Complete clustering method to yield 6 clusters which were further assembled into profiles Suppression, Prevention, Up and Down. Clustering distance, clustering method and number of clusters were selected so that all meaningful clusters were identified by the analysis. For heatmaps rlog counts from DESeq2 analysis were scaled by row (row Z-score) and low to high expression levels are indicated by blue-white-red color gradient. 2 representative biological replicates are shown for each sample in the figures for visualization purposes, count data from all replicates are available in the S6 Table.

### Chromatin immunoprecipitation (ChIP)

For the ChIP with formaldehyde crosslinking, macrophages (3–10 x $10^6$ cells per condition, 2–4 biological replicates/ macrophage donors per condition, see S17 Table) were washed once with warm PBS and incubated for 30 min at 37°C with accutase (eBioscience) to detach the cells. ChIP protocol steps were performed as described in [122], except that BSA-blocked ChIP grade protein A/G magnetic beads (Thermo Fisher Scientific) were added to the chromatin and antibody mixture and incubated for 2 h at 4°C rotating to bind chromatin-antibody

complexes. Samples were incubated for ~3 min with a magnetic stand to ensure attachment of beads to the magnet and mixed by pipetting during the wash steps. Eluted DNA was either subjected to ChIP-seq library preparation or used for ChIP-qPCR experiments. Input chromatin DNA was prepared from 1/4 of chromatin amount used for ChIP. Antibodies used for ChIP were anti-H3K4me3 (Merck Millipore, 04–745, 4 µl per ChIP), anti-H3K27me3 (Merck Millipore, 07–449, 4 µl per ChIP), anti-H3K27ac (abcam, ab4729, 4 µg per ChIP) and anti-H3K4me1 (Cell Signaling, 5326S, 5 µl per ChIP).

## ChIP library preparation and sequencing

ChIP-seq libraries were constructed with 1–10 ng of ChIP DNA or input control as a starting material. Libraries were generated using the NEXTflex ChIP-Seq Kit (Bioo Scientific) as per manufacturer's recommendations. Concentrations of all samples were measured with a Qubit Fluorometer (Thermo Fisher Scientific) and fragment length distribution of the final libraries was analyzed with the DNA High Sensitivity Chip on an Agilent 2100 Bioanalyzer (Agilent Technologies). All samples were normalized to 2 nM and pooled equimolar. The library pool was sequenced on the NextSeq500 (Illumina) with 1 x 75 bp and total 18.6 to 41 million reads per sample.

## ChIP-seq analysis

Sequencing reads containing bases with low quality scores (quality Phred score cutoff 20) or adapters were trimmed using TrimGalore program (http://www.bioinformatics.babraham.ac.uk/projects/trim_galore/). BWA [123] was used to align reads from FASTQ files to hg19 human reference genome. Samtools [124] was used for manipulations (e.g. sorting, indexing, conversions) of the sequencing files. Picard (https://broadinstitute.github.io/picard/) was used for duplicate read removal. BEDTools [125] was used for generation of BED files. The following command was used for the alignment to hg19:

*bwa mem -M <reference genome> <ChIP-seqfile.fastq> | samtools view -bT <reference genome> | samtools view -b -q 30 -F 4 -F 256 > <aligned-ChIP-seq-file.bam>*

ChIP-seq data have been deposited in the ArrayExpress database at EMBL-EBI (www.ebi.ac.uk/arrayexpress) under accession number E-MTAB-10475.

## ChIP-seq peak calling

MACS2 peak calling [126] against the input control was used for H3K4me3 and H3K27ac with *-q 0.01* parameter. SICER peak calling [127] against the input control was used for H3K4me1 and H3K27me3 with default settings. SICER peaks for ChIP enrichment over background with fold change ≥ 2 were selected. MACS2 and SICER peaks were filtered to exclude blacklisted regions [128]. To generate a file with all peaks for each modification, peaks found in replicates for different conditions were pooled together and merged using BEDTools[125].

## ChIP-seq differential region analysis

DiffReps [129] was used for identification of differentially enriched regions/ dynamic regions between all collected biological replicates (see S17 Table) from mock, WAC, WA314, WA314ΔYopP and WA314ΔYopM. Csaw [130] normalization coefficients for efficiency bias were calculated with *width 150*, *spacing 100*, *minq 50*, *ext 210* and used as input normalization coefficients in diffReps. Other diffReps parameters were set to default except for H3K4me1 and H3K27me3 where–*nsd broad* parameter was used. Accuracy of differentially enriched regions was examined in IGV [131]. Statistically significant sites with log2 FC ≥1 and adjusted

P-value ≤0.05 were selected for further analysis. Regions were further filtered to exclude regions that do not overlap MACS2 or SICER peaks, overlap blacklist regions [128] and regions with low read counts (less than 20 for any replicate in the upregulated condition; less than 20 in the upregulated condition for H3K4me3). Additionally, H3K27me3 regions with length less than 1700 bp were excluded as smaller regions appeared to be false positives. For further clustering analysis of H3K4me3 and H3K27ac dynamic regions at promoters and enhancers all differential sites for each mark for different conditions were pooled together and merged.

### ChIP-seq quantification of tag counts

For quantification of tag counts at peaks and differential sites raw counts at target regions were quantified using EaSeq [132] and normalized with csaw [130] efficiency bias coefficients with the formula: *(1+raw counts)/(csaw effective library size/$10^6$)/bp per kb*.

### ChIP-seq classification and annotation of regions

Promoter coordinates of ±2 kb from TSS and associated gene annotations were extracted from RefSeq hg19 gene annotations in UCSC Genome Browser [133] using EaSeq [132]. Regions overlapping promoter coordinates were defined as regions at promoters and annotated with associated genes; one region could overlap multiple promoters and associate with multiple genes. Regions that did not overlap promoters but overlapped with H3K4me1 regions were assigned to enhancers. Enhancer regions were annotated to the closest gene in EaSeq. The remaining regions were classified as "undefined".

For the identification of "dynamic" and "constant" peak regions, pooled peaks (based on MACS2 or SICER) were intersected with pooled differential site regions (based on diffReps) for each histone modification. Peaks intersecting differential regions were defined as "dynamic", whereas the rest of the peaks were termed "constant".

Latent enhancers were defined as sites outside promoters with H3K4me1 signal increase based on diffReps without pre-existing H3K4me1 and H3K27ac signal (no SICER and MACS2 peaks).

### ChIP-seq clustering of promoters and enhancer regions

Clustering of normalized tag counts at pooled differential sites was performed with R function heatmap.2. All H3K4me3 and H3K27ac regions from pooled differential sites that intersected promoter regions were used for promoter (P1a-P2d classes) heatmaps. For the H3K27ac enhancer heatmap (E1-E4 classes) all H3K27ac regions from pooled differential sites outside promoters and overlapping H3K4me1 peaks were used. H3K4me3 promoter heatmap was generated with clustering distance based on Pearson correlation and Complete clustering method. For H3K27ac promoter heatmap regions which did not overlap regions in H3K4me3 promoter heatmap were used for clustering. H3K27ac promoter and enhancer heatmaps were generated with clustering distance based on Spearman correlation and Average clustering method. Clustering distance, clustering method and number of clusters were selected so that all meaningful clusters were identified by the analysis. Clusters were further assembled into classes based on the pattern of histone modifications across conditions.

### ChIP-seq comparison to public data

For comparison of H3K27ac data from this study and publicly available data [10,64] raw tag counts at all H3K27ac peaks from this study were normalized with csaw [130] efficiency bias

coefficients. Batch effect between different datasets was removed using limma package [121] and tag counts were used to calculate Spearman correlation.

## ChIP-qPCR

ChIP-qPCR was performed with SYBR Green/ROX qPCR Master Mix kit (Thermo Fisher Scientific) following manufacturer's instructions. Primers (S15 Table) were designed using PRIMER-Blast tool [134] with the optimal melting temperature of 60˚C and template length between 55 and 200 bp. For all primer pairs input chromatin DNA was used to generate standard curves and verify amplification efficiency between 90–100%. The specificity of primers was confirmed using reaction without a DNA template and melting curve analysis of PCR products. qPCR was performed on a Rotorgene 6000 qPCR machine (Qiagen) and analyzed with the Rotor-Gene 6000 software (Qiagen). A gain optimization was carried out at the beginning of the run. Concentration of amplified DNA was calculated based on the standard curve. In order to compare changes in enrichment at specific regions between different conditions, normalization was done with at least 2 selected control regions which did not show change in histone modifications during infection.

## RT-qPCR analysis

Uninfected or infected primary human macrophages were subjected to RNA extraction using RNeasy Mini Kit (Qiagen) following manufacturer's instructions. 1 μg of each RNA was reverse transcribed using iScript cDNA Synthesis Kit (Bio-Rad Hercules) and subjected to RT-qPCR reaction using the TaqMan Fast Advanced Master mix (Applied Biosystems) or SYBR Green/ROX qPCR Master Mix kit (Thermo Fisher Scientific) and gene specific primers (see S15 Table for primers).

For determination of IL1B and TNF mRNA levels the TaqMan Gene Expression Assays for human IL1B (Hs00174097_m1) and TNF (Hs01113624_g1) were employed. As reference the TagMan Gene Expression Assay for GAPDH (Hs02758991_g1), TATA-box binding protein (TBP) (Hs00427620_m1) and beta-2-microglobulin (B2M) (Hs00187842_m1) were used (all from Thermo Fisher Scientific). PCR was performed using the LightCycler 480 Instrument (Roche Life Science) and data were analyzed according to manufacturer's instruction (Roche LightCycler 480 software; Software release 1.5.1.62). Reference genes and external standards were employed for the relative quantification of gene expression.

For determination of PPARGC1B and ABI1 mRNA levels RT-qPCR was performed with SYBR Green/ROX qPCR Master Mix kit (Thermo Fisher Scientific) following manufacturer's instructions. Primers (S15 Table) were designed using PRIMER-Blast tool [134] with the optimal melting temperature of 60˚C, template length between 55 and 200 bp and including intron junctions. cDNA was used to generate standard curves and verify amplification efficiency between 90–100%. The specificity of primers was confirmed using reaction without a cDNA template or reverse transcription reaction without the reverse transcriptase and melting curve analysis of PCR products. qPCR was performed on a Rotorgene 6000 qPCR machine (Qiagen) and analyzed with the Rotor-Gene 6000 software (Qiagen). A gain optimization was carried out at the beginning of the run. Concentration of amplified cDNA was calculated based on the standard curve and normalized using reference genes GAPDH and HPRT.

## Pathway analysis

Gene Ontology (GO) and Kyoto Encyclopedia of Genes and Genomes (KEGG) terms were determined for RNA-seq and ChIP-seq gene lists by using DAVID webtool [135,136].

## Motif analysis

Transcription factor (TF) motif enrichment for known motifs was performed using HOMER package [137]. Command *findMotifsGenome.pl* was used and a list of genomic coordinates was supplied as an input; the exact size of supplied regions was used by setting parameter *-size given*.

## Boxplots

Boxplots were generated using ggplot2 in R. Boxes encompass the twenty-fifth to seventy-fifth percentile changes. Whiskers extend to the tenth and ninetieth percentiles. Outliers are depicted with black dots. The central horizontal bar indicates the median.

## Association between RNA-seq and ChIP-seq

Gene lists from RNA-seq and ChIP-seq were compared based on gene symbols to find the number of overlapping genes and determine how many genes showed associated epigenetic and gene expression changes. In some cases RNA-seq lists contained old gene symbols which did not match to symbols in ChIP-seq lists with RefSeq annotation from UCSC. All old symbols in RNA-seq lists without a match in RefSeq annotation were replaced with a new symbol using Multi-symbol checker tool [138].

Relative overlap between promoter and enhancer classes was obtained by dividing the number of overlapping genes from two input classes with the total number of genes from the first input class and then with total number of genes from the second input class.

## Calculation of % YopP effect

Percentage YopP effect was calculated as ratio of FC between WA314ΔYopP vs WA314 and WAC vs WA314 (Suppression and Prevention) or WA314 vs mock (Up and Down). % YopP effect was presented for individual strongly WA314-regulated genes and regions from inflammatory response and Rho GTPase pathway, which associated with gene expression and histone modification changes in Suppression and Prevention profiles. Specifically, only genes were selected, which overlapped WAC vs WA314 DEGs and differentially enriched regions. Normalized tag counts were calculated for the target regions and used to obtain FC.

## Rho GTPase pathway gene analysis

The target gene list with 534 Rho GTPase pathway genes was compiled from publicly available data and included 370 effectors binding GTP-Rho GTPases [87,91,92], 66 GAPs, 77 GEFs [90] and 23 Rho GTPases [93,94]. The list of genes does not match the list including activities (534 vs 536) as some genes possess multiple activities.

## Immunofluorescence staining

For analysis of YopP effect on actin and ABI1 infected cells were washed twice with PBS, fixed with 4% PFA in PBS for 5 min and permeabilized with 0.1% Triton X-100 (w/v) in PBS for 10 min. After fixation and permeabilization coverslips were washed twice with PBS. Unspecific binding sites were blocked with 3% bovine serum albumin (BSA, w/v) in PBS for at least 30 min. Samples were then incubated with 1:100 (anti-ABI1 (Sigma-Aldrich), anti-ZO-1 (Zymed), anti-H3K27ac (Abcam)) or 1:200 (anti-vinculin (Sigmal-Aldrich)) dilution of the primary antibody for 1 h and incubated with a 1:200 dilution of the suitable fluorophore-coupled secondary antibody for 45 min. Secondary anti-IgG antibodies and dyes used: Alexa488

chicken anti-rabbit (Molecular Probes), Alexa568 goat anti-mouse (Molecular Probes), Alexa-Fluor568 phalloidin (Invitrogen, GIBCO), AlexaFluor633 phalloidin (Invitrogen, GIBCO).

After each staining coverslips were washed three times with PBS. Both primary and secondary antibodies were applied in PBS supplemented with 3% BSA. Fluorophore-coupled phalloidin (1:200, Invitrogen) was added to the secondary antibody staining solution as indicated. Coverslips were mounted in ProLong Diamond (Thermo Fisher Scientific).

For analysis of cells positive for effector translocation primary human macrophages were infected with different MOIs (10, 25, 50 and 100) of *Y. enterocolitica* WA314-YopE-ALFA strain for 6 h and fixed with 4% PFA for 10 min. The samples were permeabilized with 0.1% Triton-X-100 for 15 min, blocked with 3% BSA for 1h and incubated with anti-ALFA nanobody-635 (FluoTag-X2 anti-ALFA Abberior Star 635P) (1:300) for 1h at RT. After washing 3 times with PBS, cells were counterstained with DAPI and phalloidin- 488 and mounted on glass slides in Prolong Glass Antifade Mountant (Thermo Fisher Scientific).

## Microscopy

Fixed samples were analyzed with a confocal laser scanning microscope (Leica TCS SP8) equipped with a 63x, NA1.4 oil immersion objective and Leica LAS X SP8 software (Leica Microsystems) was used for acquisition.

## Image analysis

For analysis of YopP effect on actin and ABI1 the z-stacks of images were combined to one image using maximum intensity projection. These images were used to measure the mean fluorescent intensity of the F-actin signal with a circle (9.02 μm diameter) around the cellular junctions between macrophages. The signal intensity at junctions refers to the maximum value of 255 for 8-bit images.

## Western blot analysis

Primary human macrophages ($1–2*10^6$ cells per condition) were washed once with warm PBS and scrapped off the plates to harvest the cells. Cell were pelleted by centrifugation at 700 x g, 10 min and 4˚C. For ABI1 western blot cells were resuspended in 60 μl Lysis buffer (50 mM HEPES-KOH, pH 7.5, 140 mM NaCl, 1 mM EDTA, 10% glycerol, 0.5% NP-40, 0.75% Triton X-100, protease inhibitor (Complete, Roche Diagnostics)) and lyzed by incubation at 4˚C for 30 min. For caspase-3 western blot cells were resuspended in 40 μl RIPA buffer (50 mM Tris, pH 8.0, 150 mM NaCl, 1% TRITON-X-100, 0.5% sodium deoxycholate, 0.1% SDS, protease inhibitor (Complete, Roche Diagnostics)) and lyzed by incubation at 4˚C for 10 min and brief sonication. Lysates were centrifuged at 10000 x g, 10 min and 4˚C and supernatant was collected for western blot analysis.

Proteins were separated by SDS-PAGE and transferred to polyvinylidene difluoride (PVDF) membrane (Immobilon-P, Millipore) by semi-dry blotting. The membrane was incubated in blocking solution (5% milk powder (w/v) for ABI1 or 3% BSA (w/v) for GAPDH and caspase-3 in TBS supplemented with 0.03% Tween 20; TBS-T) at room temperature for 30 min. Primary antibody incubations (anti-ABI1 (Sigma-Aldrich) at 1:1000, anti-GAPDH (Sigma-Aldrich) at 1:3000, anti-caspase-3 (Cell Signaling) at 1:1000) were carried out at 4˚C overnight, secondary antibody incubation (horseradish peroxidase linked anti-rabbit IgG (Cell Signaling) at 1:10000) was performed at room temperature for 1 h. Washing steps with TBS-T were done between the incubations. Antibody signals were visualized with chemiluminescence technology (Supersignal West Femto, Pierce Chemical) and captured on X-ray films (Fujifilm) or digital imager (Licor).

## Supporting information

**S1 Fig. Analysis of infection conditions. A,** Left: Immunofluorescence staining of primary human macrophages mock infected or infected with WA314-YopE-ALFA for 6 h with MOI 10, 25, 50 or 100. Cells were stained with anti-ALFA nanobody-635 (red) to visualize YopE, Alexa568 phalloidin (green) to visualize actin and DAPI (blue) to visualize nucleus. Right: Quantification of YopE-ALFA positive cells. Bars show mean and error bars represent standard deviation when analyzing two different coverslips. **B,** Line plot showing time course H3K4me3 (IL1B, SOCS3, IDO1 genes) and H3K27ac (IL6, SOCS3 genes) ChIP-qPCR analysis of WAC infected primary human macrophages with MOI of 100. Lines represent means and error bars represent standard deviation from two independent macrophage donors/ biological replicates. **C,** Bar plot showing membrane permeability due to cell death in primary human macrophages mock infected, infected with WA314 (MOI of 100) or treated with positive (pos.) control (1% TRITON-X-100) for 6 h. Increased membrane permeability is expressed as higher SYTOX Green intensity, where the dye binds to DNA and emits fluorescence when plasma membrane integrity is compromised. Bars represent means of two biological replicates/ macrophage donors (dots with different shapes). **D,** Western blot analysis of primary human macrophages showing caspase-3 cleavage in mock or WA314 (MOI 100) infected cells or treated with positive control Staurosporine (5 μM) for 6 h. * indicates full length caspase-3 (35 kDa) and cleavage fragments (17 and 19 kDa). GAPDH was used as a loading control. Data are representative of four biological replicates/ different macrophage donors. **E,** Bar plot showing colony forming unit (CFU) analysis of *Y. enterocolitica* WA314 growth during infection of primary human macrophages for 0 h, 1.5 h or 6 h at 37˚C. Bars represent means of two different biological replicates (dots with different shapes). **F,** Flow cytometry analysis of monocyte-derived primary human macrophages after 6 days of differentiation. Live cells (L/D negative) were selected (first top panel) and doublets excluded (second top panel). CD14 positive macrophages were selected (third top panel) and analyzed for expression of macrophage markers CD163, CD38, CD86, HLA-DR,CD206 and CD16 (bottom panels).
(TIF)

**S2 Fig. Confirmation of ChIP-seq and RNA-seq results by qPCR. A,** Bar plot showing H3K27me3 (left) and H4K4me3 (right) ChIP-qPCR from primary human macrophages mock infected or infected with WAC or WA314 for 6 h with MOI of 100. HOXC12 and NXPH4 are positive controls for H3K27me3 and ADH5 and TUBA1C are positive controls for H3K4me3. Data are representative of at least two experiments with different biological replicates/ macrophage donors. **B,** Peak tracks showing tag density of H3K4me3 (red) or H3K27ac (blue) ChIP-seq from primary human macrophages infected for 6 h with indicated strains (MOI of 100) (left). qPCR target refers to the site used for fragment amplification in ChIP-qPCR analysis shown as a dot plot (right). The ChIP-qPCR signal was expressed as relative (rel.) enrichment (enr.) vs mock. Lines represent means from at least two different biological replicates/ macrophage donors (dots with different shapes). **C,** Dot plots showing gene expression analysis of PPARGC1B, ABI1, TNF and IL1B genes with RT-qPCR from primary human macrophages infected for 6 h with indicated strains (MOI 100). The RT-qPCR signal was expressed as fold vs mock. Lines represent means from at least two different biological replicates/ macrophage donors (dots with different shapes).
(TIF)

**S3 Fig. Analysis of association between histone marks and gene expression. A-D,** Bar plots showing fraction (%) of genes associated with H3K4me3 at promoter **(A)**, H3K27ac at promoter **(B)**, H3K4me3 and H3K27ac at promoter **(C)** and H3K27ac at enhancer **(D)** changes

for comparisons between mock, WAC and WA314 showing significant ($\geq$ 2-fold change, adjusted P-value $\leq$ 0.05) associated change in gene expression (overlap RNA).
(TIF)

**S4 Fig. Pathway and transcription factor motif enrichment analysis of genes with associated histone modification and gene expression changes during *Y. enterocolitica* infection. A,** Heatmap of row-scaled (row Z-score) RNA-seq rlog gene counts for genes with RNA-seq and ChIP-seq promoter or enhancer overlaps belonging to "Inflammatory response" pathway as in Fig 3H. **B,** GO terms associated with latent enhancer genes from Fig 2F. **C-D,** Heatmaps of row-scaled (row Z-score) RNA-seq rlog gene counts for selected genes with RNA-seq and ChIP-seq promoter or enhancer overlaps belonging to phosphatases and kinases from MAPK signaling in Suppression profile **(C)** and Rab GTPase pathway in Prevention and Suppression profiles **(D)** from S16 Table. **E,** Heatmap showing transcription factor motif enrichment in promoter and enhancer regions of genes with RNA-seq and ChIP-seq overlaps from Fig 3G. Color indicates the level of log10 transformed P-value positively correlating with significance of enrichment.
(TIF)

**S5 Fig. Epigenetic and transcriptional regulation of Rho GTPase pathway genes in *Y. enterocolitica* infected macrophages. A,** Bar plot showing fraction and number (in bars) of indicated Rho GTPase pathway genes with changes in expression only (RNA only), histone modification only (ChIP only), overlaps between RNA-seq and ChIP-seq (RNA & ChIP) and no overlap with neither ChIP-seq nor RNA-seq (no overlap). **B,** Bar plot showing number of Rho GTPase pathway genes with dynamic histone modifications at promoters, enhancers or both (overlap). **C,** Table showing number of Rho GTPase pathway genes with histone modification changes at enhancers and number of associated enhancers. "% total enhancer associated genes" indicates what fraction of all Rho GTPase pathway genes with changes at enhancers show a certain number of changed enhancers. **D, E,** Heatmaps of DEGs with associated histone modification changes at promoters or enhancers encoding Rho GTPase effectors **(D)** and GEFs **(E)**. Associated classes are color coded on the left. The specificity of GEFs for Rho GTPases is color coded on the right **(E)**. Gene rlog counts were row-scaled (row Z-score). **F,** Table showing specificity of Rho GTPase effectors associated with RNA-seq and ChIP-seq overlaps in **(A)** and Fig 4C.
(TIF)

**S6 Fig. Analysis of YopM and YopP effect on epigenetic and gene expression changes in macrophages upon *Y. enterocolitica* infection. A,** Principal component analysis of H3K4me3 tag counts in classes P1a-b of macrophages infected with the indicated strains for all biological replicates used in the analysis. **B,** Percentage YopP effect (median value) for H3K4me3 (P1a-b classes) and H3K27ac (P2a-d, E1-4 classes) when compared to WA314 vs mock (Up and Down profiles) or WA314 vs WAC (Suppression and Prevention profiles). **C,** Percentage YopP effect (median value) for DEGs associated with respective histone modification changes at promoters or enhancers. "Median": median value when taking % YopP effect from all classes together. **D,** Scatter plot of % YopP effect for RNA-seq DEGs and ChIP-seq differential regions for genes with associated gene expression and histone modification change from Rho GTPase and Inflammatory response pathways from Fig 5C and 5D. Purple dots associated with ABI1 gene are indicated. **E,** Top: Dot plots showing ChIP-qPCR of H3K27ac at enhancer within ABI1 gene and H3K4me3 at ABI1 promoter from primary human macrophages infected for 6 h with indicated strains (MOI of 100). Lines represent means from three different biological replicates/ macrophage donors (dots with different shapes). Bottom: Peak tracks showing tag

density of H3K27ac at enhancer (blue) and H3K4me3 at promoter (red) from ChIP-seq and site used for fragment amplification in ChIP-qPCR analysis (qPCR target) shown in dot plots. The ChIP-qPCR signal was expressed as relative (rel.) enrichment (enr.) vs mock. inh: MAPK inhibitors. **F,** Dot plots showing RT-qPCR gene expression analysis of ABI1 from primary human macrophages infected for 6 h with indicated strains (MOI of 100). The RT-qPCR signal was expressed as fold vs mock. Lines represent means from three different biological replicates/ macrophage donors (dots with different shapes). inh: MAPK inhibitors. **G,** Immunofluorescence staining of primary human macrophages infected with WA314ΔYopP+C646 (p300 inhibitor) or WA314ΔYopP with MOI of 100 for 6 h. Cells were stained with antibody for H3K27ac. Images are representative of two independent experiments.
(TIF)

**S1 Table.** *Yersinia enterocolitica* **strains used in the study.**
(XLSX)

**S2 Table. Analysis of promoter regions in P1 and P2 modules.**
(XLSX)

**S3 Table. Analysis of enhancer regions in E1-4 classes.**
(XLSX)

**S4 Table. Analysis of latent enhancers.**
(XLSX)

**S5 Table. RNA-seq background from DESeq2 analysis.**
(XLSX)

**S6 Table. Clustering of RNA-seq DEGs in a heatmap.**
(XLSX)

**S7 Table. Metabolic pathways in Prevention profile.**
(XLSX)

**S8 Table. Enriched pathways associated with Rho GTPase pathway genes.**
(XLSX)

**S9 Table. Analysis of Rho GTPase pathway genes in promoter and enhancer classes.**
(XLSX)

**S10 Table. RNA-seq heatmap of DEGs from Rho GTPase pathway.**
(XLSX)

**S11 Table. Rho GTPase specificity of GAPs and GEFs with RNA-seq and ChIP-seq overlaps in different profiles.**
(XLSX)

**S12 Table. Rho GTPase specificity and reported function in epigenetics for effectors with RNA-seq and ChIP-seq overlaps.**
(XLSX)

**S13 Table. Analysis of YopP effect on Rho GTPase pathway genes.**
(XLSX)

**S14 Table. Analysis of YopP effect on inflammatory response pathway genes.**
(XLSX)

**S15 Table. Primer sequences used for ChIP-qPCR and RT-qPCR analysis.**
(XLSX)

**S16 Table. Enriched pathways associated with trafficking, MAPK signaling, Akt signaling and cell death.**
(XLSX)

**S17 Table. Summary of all replicates used for ChIP-seq and RNA-seq experiments.**
(XLSX)

**S18 Table. All filtered differentially enriched regions for histone marks based on diffReps analysis.**
(XLSX)

## Acknowledgments

We thank the UKE Core Facility Microscopy Imaging (Umif) for help with experimental set-ups and data analysis. We also would like to thank Daniela Indenbirken for generating the RNA-seq libraries and Frank Bentzien (UKE Transfusion Medicine) for buffy coats. We thank Alexander Carsten and Asiri Ortega for WA314-YopE-ALFA strain, Sahaja Aigal for WA314-YopE-ALFA stainings and Eva Tolosa and Manuela Kolster for flow cytometry analysis.

## Author Contributions

**Conceptualization:** Indra Bekere, Marie Schnapp, Laura Berneking, Thomas Günther, Martin Aepfelbacher.

**Data curation:** Indra Bekere, Jiabin Huang.

**Formal analysis:** Indra Bekere, Jiabin Huang, Marie Schnapp, Maren Rudolph, Laura Berneking.

**Funding acquisition:** Adam Grundhoff, Nicole Fischer, Martin Aepfelbacher.

**Investigation:** Indra Bekere, Jiabin Huang, Marie Schnapp, Maren Rudolph, Laura Berneking.

**Methodology:** Indra Bekere, Jiabin Huang, Marie Schnapp, Maren Rudolph, Laura Berneking, Thomas Günther.

**Project administration:** Martin Aepfelbacher.

**Resources:** Martin Aepfelbacher.

**Supervision:** Martin Aepfelbacher.

**Validation:** Marie Schnapp, Laura Berneking.

**Visualization:** Indra Bekere, Maren Rudolph.

**Writing – original draft:** Indra Bekere, Martin Aepfelbacher.

**Writing – review & editing:** Indra Bekere, Klaus Ruckdeschel, Adam Grundhoff, Thomas Günther, Nicole Fischer.

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
