## [Decision Letter · Decision Letter 0]

20 Aug 2021

Dear Ms Bekere,

Thank you very much for submitting your manuscript "Epigenetic reprogramming of human macrophages by an intestinal pathogen" for consideration at PLOS Pathogens. As with all papers reviewed by the journal, your manuscript was reviewed by members of the editorial board and by several independent reviewers. In light of the reviews (below this email), we would like to invite the resubmission of a significantly-revised version that takes into account the reviewers' comments.

As suggested by reviewer 1 and alluded to by other reviewers, I would suggest major revisions in data presentation and contextual revisions are needed across manuscript.

Please demonstrate the maturity and purity of the macrophages that you used.

I agree with reviewer 2 regarding the problems associated with analyses being based on two biological replicates, please address with additional replicate(s).

Please include validation of detected histone modification marks and gene expression.

Please address reviewer 3’s concern regarding evidence that histone modification is responsible for differential ABI1 gene expression and linked to actin cytoskeleton organization in Yersinia-infected macrophages.

We cannot make any decision about publication until we have seen the revised manuscript and your response to the reviewers' comments. Your revised manuscript is also likely to be sent to reviewers for further evaluation.

Sincerely,

Denise M. Monack

Section Editor

PLOS Pathogens

Denise Monack

Section Editor

PLOS Pathogens

Kasturi Haldar

Editor-in-Chief

PLOS Pathogens

orcid.org/0000-0001-5065-158X

Michael Malim

Editor-in-Chief

PLOS Pathogens

orcid.org/0000-0002-7699-2064

As suggested by reviewer 1 and alluded to by other reviewers, I would suggest major revisions in data presentation and contextual revisions are needed across manuscript.

Please demonstrate the maturity and purity of the macrophages that you used.

I agree with reviewer 2 regarding the problems associated with analyses being based on two biological replicates, please address with additional replicate(s).

Please include validation of detected histone modification marks and gene expression.

Please address reviewer 3’s concern regarding evidence that histone modification is responsible for differential ABI1 gene expression and linked to actin cytoskeleton organization in Yersinia-infected macrophages.

Reviewer's Responses to Questions

**Part I - Summary**

Reviewer #1: In the manuscript entitled Epigenetic reprogramming of human macrophages by an intestinal pathogen, the authors provide a large amount of data on histone modifications induced by Yersinia enterocolitica during infection of human macrophages. The data presented here include genome wide ChIP-seq data on 4 different histone marks under 4 different conditions, as well as expression data for the 4 different conditions. In general, the data is of interest and constitutes a valuable resource, however the way in which the data is presented and the organization of the manuscript hinder the biological messages and take away from the significance of the results. As it stands, the paper is difficult to read, the figures are confusing, and the biological significance is lost and hard to follow. Therefore, major revisions in data presentation and contextual revisions are needed across text and methods.

Reviewer #2: The present study represents a global analysis of epigenetic histone modifications in primary human macrophages by the enteropathogenic pathogen Yersinia enterocolitica using genome-wide chromatin immunoprecipitation (ChIP) sequencing (CHIP-seq) which is compared with a parallel transcriptomic analysis (RNA-seq). The authors document that Y. enterocolitica infections are associated with major alterations of the chromatin state (H3K4me1, H3K4me3 and H3K27ac) of macrophage gene promoters and enhancers, and that the extent of the modification is manipulated by Y. enterocolitica pathogenicity factors encoded on the Yersinia virulence plasmid. In particular, YopJ/P is implicated in the reorganization of epigenetic histone modifications and associated gene reprogramming. Notably, many alterations appear to influence immune responses which are associated with Rho GTPase pathways.

The analyses of the data of the experiments in this study is very complex. To illustrate and interpret the results, the authors classified them in four logical profiles: (i) ‘Suppression’ and ‘Prevention’ for Yersinia-induced deposition or removal of histone marks and (ii) ‘Down’ and ‘Up’ in which the virulence plasmid-encoded pathogenicity factors down- or upregulate histone marks independently of other chromosomally encoded PAMP activities. This categorization is very helpful and indicates certain but also largely varying correlations of the histone modification and the actual gene expression pattern (30% Suppression to 2% Down profile).

Reviewer #3: This manuscript describes a comprehensive analysis of epigenetic histone modification changes in human macrophages infected with wild-type Yersinia enterocolitica or corresponding mutants lacking the plasmid encoding the type III secretion system or the effector YopP. The results provide new information to the field and are interesting. I have one comment about a weakness in data presentation and interpretation and several minor editorial suggestions.

**Part II – Major Issues: Key Experiments Required for Acceptance**

Reviewer #1: Methods:

1) It would be helpful if more information was included on the infection phenotype of the macrophages that have been studied. Why was this MOI chosen? Did the authors use a gentamicin protection assay? If not, what is the bacterial growth over this period, which is likely multiple logs. Did the authors check for bacterial uptake – even though the T3SS is primed, macrophages may still internalize bacteria. What % at 6 h post-infection was internal vs. external?

This information is important as the high MOI used for a 6h infection could be problematic.

Furthermore, the authors mention that “at 6 h after infection, a time point at which a maximal effect on gene expression but no signs of apoptosis are detectable in the human macrophages”, but no data is shown on this point.

2) Please demonstrate the maturity and purity of the macrophages used in these studies. We agree that the method used from Linder et al (magnetic bead isolation by CD14) will collect monocytes, and then with serum and routine media changes to remove non-adherent cells the authors would generate macrophages. However, as stated in the methods that “Macrophages were used for infection 1-2 weeks after isolation” implies a large range of macrophage age, maturity, and responsiveness to stimulus. It would be important to show that macrophages used during this period are homogenous across time.

Contextual changes:

3) The following statement needs to be amended to include additional literature, and should be noted as it reads the authors are implying that because Yersinia replication extracellularly in draining lymph nodes (a hallmark of disease) the organism may suppress innate immune functions (etc…). This is not an accurate description, as part of the pathogenesis cycle does include intracellular niche depending upon the route of colonization of the host for all Yersinia spp. Perhaps it is more prudent to subjectively mention this with appropriate citations when applicable (St. John, Pujol, Hinnebusch…).

“Pathogenic Yersinia species, which comprise the entero-pathogens Y. pseudotuberculosis and Y. enterocolitica as well as the plague agent Yersinia pestis, proliferate extracellularly in lymphoid tissues of animal hosts (Balada-Llasat & Mecsas, 2006; Viboud & Bliska, 2005). Therefore, the bacteria suppress phagocytosis, migration and immune signaling in resident cells of the innate immune system (Viboud & Bliska, 2005; Marketon et al., 2005).”

4) This statement should be rephrased. As written implies it is only from the T3SS, there are also LPS, adhesions, YAPs, etc.. that the host cell may also interact with/sense during infection.

“Yersinia induced DAMPs are e.g. produced in response to membrane damage caused by the T3SS

translocation pore or by deactivation of Rho GTPases through YopE and YopT (Schubert et al., 2020).”

5) “Previous systematic studies of Yersinia effects on gene expression were conducted with gene arrays and cultured mouse macrophages infected for up to 2.5 h (Sauvonnet et al., 2002; Hoffmann et al., 2004).”

Again, the authors should include the studies of other Yersinia spp – one of which was an RNAi genome-wide screen in mouse macrophages to identify exploited pathways required for disease. Suggestion is to mention this with appropriate citations when applicable (Meagher, Sjostedt, Connor, Jett, Dersch, Bliska…).

Organization of results:

6) As it is presented the authors focus mainly on the epigenetic changes without extracting the biological significance, which is present and interesting. A reorganization of the results and better contextualization of them would help immensely. The suggestion would be to organize the manuscript around the biological findings (already present in the text) cited below:

- WA314 suppresses PAMP-LPS-induced histone modifications.

All the data that illustrates this point should be grouped in one section (including promoters and enhancers), and functional analysis should be added. Indeed, in the context of the already known signaling suppression by Yersinia, it would be interesting to see the changes at the chromatin level, implying more than just suppression of gene activation.

- YopP but not YopM contributes to epigenetic changes produced by WA314.

Again, the most of the data is there, the authors need to spend more time on the functional analysis of the genomic regions differentially regulated by these two factors.

- Epigenetic modifications do not correlate well with transcriptional changes.

This message is not a problem but more an interesting finding. Indeed, this shows that beyond gene expression, there are “silent” chromatin modifications that could have a large impact on other physiological outcomes than infection. In line with this, the message that chromatin modifications correlate with expression should be removed, along with figure 4J. In this section the genes could be sub-classified to give biological meaning and importance to the genes in each category.

o Upregulated genes in WAC or WAC13

o Upregulated genes that are associated with HM at promoters

o Upregulated genes that are associated with HM at enhancers

o Upregulated genes that are associated with HM at promoters and enhancers

- Reprogramming of the Rho GTPase pathway.

Could the analysis be taken one step further to determine if a common transcription factor could be regulating all the genes in this category?

7) In general there is a substantial amount of Yersinia work dedicated to signaling (MAPK, PI3K/AKT, Caspase(s) etc…) and trafficking (Rabs, VAMPs, SNAREs etc..) pathways exploited by this bacteria for pathogenesis that is not discussed in terms of this chromatin work. Include this contextual knowledge would strengthen the manuscript across all domains of Yersinia pathogenesis research, and in the context of what/how bacterial induced chromatin remodeling may interplay with these findings.

8) The terminology of the “suppression”, “prevention” profiles are very confusing and should be replaced or removed. Similarly, the scheme in figure 2B should be removed as it is confusing.

Data interpretation:

There are many general conclusions that are drawn from the results that are overinterpreted.

9) The title should be revised. For example, “Chromatin reorganization in macrophages upon Y. enterocolitica infection” or a more specific biological question. “Epigenetic reprogramming” is too strong of a statement here.

10) In the abstract and author summary: “Y enterocolitica profoundly reorganizes HM to reprogram key gene expression of the innate immune response”. This statement makes it sound as if Yersinia actively manipulates the host for HM. However, the authors describe in the manuscript how a big part of the response is common to LPS exposure. Also, chromatin reorganization could be a response of macrophages to Y. enterocolitica, not at active reprograming of the bacteria to the host. This statement is mentioned again at the end of section 1 and in the discussion.

11) In the introduction: “Histone modifications determine DNA accessibility and transcriptional programs”. Determine is a very strong word. Influence is a better word, as the correlation between HM and transcription is not black or white. Also, in the introduction H3K4me3 is referred as a promoter mark but this mark also gets enriched in gene bodies of active genes.

Reviewer #2: 1. Although the overall sets of data are largely descriptive, they give important information and reveal a new level of complexity of Yersinia-triggered macrophage responses which are of great interest for the community. However, from what I understood from their rather cryptic description about the quantitative set-up (biological and technical replicates) of the experiment and analyses, the entire set of analyses is based on two biological replicates which makes a statistical analysis and statements about the significance of identified histone modification and gene expression changes rather difficult.

In the material method section, they state that for RNA-seq analysis two representative replicates are shown for each sample (p. 32 last line), but it is unclear how many biological and technical replicates were included into the entire analysis, and which were included into the analysis of differential expression with DESeq2. A more detailed description of the biological and technical replicates used for the histone modification and gene expression analyses (statistical analysis - p values) is required to allow a proper judgment for the significance of the data. This seems particularly important as histone modification data and expression data between the documented two representative data sets seem to vary considerably with respect to several illustrated responses (Fig. 4A for WAC and WA314; Fig. 5D: WAC positive regulation, WA314 Wnt signalling, Fig, 5E WA314; Fig. 6B and 6E for WA314, Fig, S4 E WAC and WA314 etc.).

2. In the figures, exact numbers of genes, promoters and enhancers which are modified with respect to histones and/or gene expression upon the infection with applied Yersinia strains are given. However, (in line with the statistical analysis) the validity of the data is unclear. Have the authors performed assays to validate the detected histone modification marks and the gene expression data, which is important for the reliability of the data, e.g. proof of principle experiments for the Rho proteins?

Reviewer #3: Page 24, the ABI1 protein western blot data (Fig. S5E) should be move to a main figure and some quantification should be provided to confirm the differences are reproducible. At a minimum the authors should show the results of the two experiments and show some semi-quantification.

End of Page 24, what is the evidence "that the histone modification-driven reprogramming of ABI1 gene expression in fact can alter actin cytoskeleton organization in Yersinia infected macrophages"? The authors need to change this conclusion or provide evidence that histone modification is responsible for differential ABI1 gene expression.

**Part III – Minor Issues: Editorial and Data Presentation Modifications**

Reviewer #1: 1) Please number the lines as this makes reviewing easier.

2) The figure legends need to be more informative. For example it is hard to know what data was used to perform clustering figures (DEGs, genes in a specific caterogy, nearest gene to histone mark, etc.)

3) There are quite a few grammatical errors throughout which should be revised.

4) HM referring to histone mark is problematic as it can also be human macrophage – please find a new term.

5) A citation that macrophages produce TJs? This reviewer is unaware that mature macrophages made TJs.

6) The quality of the images are very bad and sometimes impossible to read.

Reviewer #2: Specific comments:

1. p. 7, introduction, 2nd paragraph: the authors state that previous systemic analyses on gene expression were performed up to 2.5 h. In the present study the authors decided to analyze histone modification and gene expression changes after 6 h. Why? Do histone modifications need this long? The authors also state that most histone modifications are transient (p.7, last sentence of the 2nd paragraph). Are the observed histone modifications maximal at 6 h? What is known about the dynamics of this process?

Moreover, are the histone modifications immediately visible in the gene expression pattern or are they somewhat delayed? If latter, could this also be one reason why observed correlation between histone modification and gene expression patterns is rather low (max. 30% for the Suppression profile). What about ChIP-seq – gene expression data from other studies - are the correlations in a similar range? These points should also be addressed/discussed.

2. p. 8, results: 1st paragraph/p. 30-34, material and methods, Fig. 1 legends: in neither of these sections, the quantitative set-up of the experiments which are basis for their overall analysis is explained. How many biological and technical replicates were used to obtain the data. Were independent samples pooled? On p. 32 last sentence states…2 representative replicates are shown for each sample. What about the other replicates? Have they included into the data sets or are the given data sets from the two selected representative samples. Why were only these two selected?

3. p. 8, l. 15, results: …H3K4me1 …. Is also found with lower levels are promoters…similar to H3K4me3, it should be added whether this indicates active or inactive promoters.

4. p. 10, l. 2-3, results: Sentence starting … ‘Overall, WAC and WA314…’ seems incomplete.

5. p. 9-12, results: This part is very descriptive and mostly lists the histone modifications (percentage) observed for the genes/promoters/enhancers in the different strains without an explanation of the consequences and could be shortened.

6. p. 13, results: 1st paragraph/p. 30-34, material and methods, Fig. 4 legends: in neither of these sections, the quantitative set-up of the experiments which are basis for their overall analysis of the gene expression patterns is explained. How many biological and technical replicates were used to obtain the data. Were independent samples pooled?

7. Table S2, S3, S5, S6, S9 please describe rows (e.g. DoA, DoB, what parameters are represented by the numbers, RPKM for RNA-seq?) and add statistical analysis (p-value etc.). The ordering by chromosome location is okay, but additional sheets in which the genes are ordered with respect to the log2fold change would be helpful.

8. p. 16, l. 14, 15: please explain abbreviation TF.

9. p. 17, l. 10: I suggest to exchange ‘there’ against ‘this’

10. p. 25, paragraph 4: the bacteria’s T3SS effectors block PAMP-associated deposition or removal – T3SS effectors should be replaced by virulence-plasmid-encoded pathogenicity factors. It is possible that other non-effector virulence factors such as YadA, which also activate MAPK pathways etc., are also implicated in reprogramming.

11. p. 26 3rd paragraph…We could exclude YopM as one of these effectors, because it did not alter any of the histone modifications – is this really true? See p. 21 3rd sentence from the buttom, in which the author state, that 31 histone modifications were differentially regulated between the yopM mutant and the wildtype.

12. p. 27-28, last and first sentence: the references added to this sentence do not document, that expression of the mentioned Rac1 associated proteins were suppressed by the bacteria…. they document that they can take over Rac1 functions, please clarify.

13. p. 30, the authors used an MOI of 100 to infect the macrophages. This is okay to obtain information about how Yersinia can manipulate human macrophages. Just for curiosity? Which MOI is expected for an in vivo infection (i.e. infections of mice, humans)? If a lower MOI is more likely, do you expect the histone modification and gene expression changes to be identical, just less pronounced, or is it likely to have a different set of histone-modified

Reviewer #3: The manuscript did not have line numbers to help with reviewing.

Page 2, change "immune cells" to "macrophages"

Page 5, "Further, YopP/J, and YopM inhibit the inflammatory responses triggered by the PAMPs or the damage associated molecular patterns (DAMPs) elicited by the bacteria.." This sentence and related text need to be revised, as the authors are equating effector-triggered immune responses, which YopM does inhibit, with DAMP-triggered immune responses.

Page 6, define "H3K27ac"

Page 19, the terminology "actually active" is awkward.

PLOS authors have the option to publish the peer review history of their article (what does this mean?). If published, this will include your full peer review and any attached files.

Reviewer #1: No

Reviewer #2: No

Reviewer #3: No
---

## [Editor Report · Decision Letter 1]

28 Oct 2021

Dear Ms Bekere,

We are pleased to inform you that your manuscript 'Yersinia remodels epigenetic histone modifications in human macrophages' has been provisionally accepted for publication in PLOS Pathogens.

Best regards,

Denise M. Monack

Section Editor

PLOS Pathogens

Denise Monack

Section Editor

PLOS Pathogens

Kasturi Haldar

Editor-in-Chief

PLOS Pathogens

orcid.org/0000-0001-5065-158X

Michael Malim

Editor-in-Chief

PLOS Pathogens

orcid.org/0000-0002-7699-2064
---

## [Editor Report · Acceptance letter]

15 Nov 2021

Dear Ms Bekere,

We are delighted to inform you that your manuscript, "Yersinia remodels epigenetic histone modifications in human macrophages," has been formally accepted for publication in PLOS Pathogens.

Best regards,

Kasturi Haldar

Editor-in-Chief

PLOS Pathogens

orcid.org/0000-0001-5065-158X

Michael Malim

Editor-in-Chief

PLOS Pathogens

orcid.org/0000-0002-7699-2064